# Functional requirements for a Samd14-capping protein complex in stress erythropoiesis

**Suhita Ray[1], Linda Chee[1], Yichao Zhou[1], Meg A Schaefer[1], Michael J Naldrett[2], Sophie Alvarez[2], Nicholas T Woods[3], Kyle J Hewitt[1]\***

[1]Department of Genetics, Cell Biology and Anatomy, University of Nebraska Medical Center, Omaha, United States; [2]Proteomics and Metabolomics Facility, University of Nebraska-Lincoln, Lincoln, United States; [3]Eppley Institute for Research in Cancer and Allied Diseases, University of Nebraska Medical Center, Omaha, United States

**Abstract** Acute anemia induces rapid expansion of erythroid precursors and accelerated differentiation to replenish erythrocytes. Paracrine signals—involving cooperation between stem cell factor (SCF)/Kit signaling and other signaling inputs—are required for the increased erythroid precursor activity in anemia. Our prior work revealed that the sterile alpha motif (SAM) domain 14 (*Samd14*) gene increases the regenerative capacity of the erythroid system in a mouse genetic model and promotes stress-dependent Kit signaling. However, the mechanism underlying Samd14's role in stress erythropoiesis is unknown. We identified a protein-protein interaction between Samd14 and the α- and β-heterodimers of the F-actin capping protein (CP) complex. Knockdown of the CP β subunit increased erythroid maturation in murine ex vivo cultures and decreased colony forming potential of stress erythroid precursors. In a genetic complementation assay for Samd14 activity, our results revealed that the Samd14-CP interaction is a determinant of erythroid precursor cell levels and function. Samd14-CP promotes SCF/Kit signaling in CD71[med] spleen erythroid precursors. Given the roles of Kit signaling in hematopoiesis and Samd14 in Kit pathway activation, this mechanism may have pathological implications in acute/chronic anemia.

**\*For correspondence:**
kyle.hewitt@unmc.edu

**Competing interest:** The authors declare that no competing interests exist.

## Editor's evaluation

This study describes a new mechanism involving Samd14 and F-actin capping protein (CP) complex formation in the regulation of stress erythropoiesis. Through orthogonal biochemical, cellular, and genetic complementation assays, the authors provide evidence that the Samd14-CP interaction is required for the proper regulation of SCF/Kit signaling in erythroid precursors in response to acute anemia. The Samd14-CP-mediated mechanism may have implications in modulating the receptor tyrosine kinase signaling in physiological and pathological conditions.

## Introduction

The expansion and differentiation potential of stem/progenitor cells in the hematopoietic system fluctuates in response to changing environmental conditions. Exposure to toxins, pathogens, blood loss, nutrient deficiency, oxygen deficits, etc. activates transcriptional programs and signal transduction pathways in progenitors and precursors to re-establish homeostasis (*Bresnick et al., 2018*; *Socolovsky, 2007*). In acute anemia, vital stress response mechanisms are communicated to erythroid progenitor and precursor cells by secreted factors, including stem cell factor (SCF), erythropoietin (Epo), bone morphogenetic protein (BMP)–4, glucocorticoids, and Hedgehog (*Bauer et al., 1999*;

**eLife digest** Anemia is a condition in which the body has a shortage of healthy red blood cells to carry enough oxygen to support its organs. A range of factors are known to cause anemia, including traumatic blood loss, toxins or nutritional deficiency. An estimated one-third of all women of reproductive age are anemic, which can cause tiredness, weakness and shortness of breath. Severe anemia drives the release of hormones and growth factors, leading to a rapid regeneration of precursor red blood cells to replenish the supply in the blood.

To understand how red blood cell regeneration is controlled, Ray et al. studied proteins involved in regenerating blood using mice in which anemia had been induced with chemicals. Previous research had shown that the protein Samd14 is produced at higher quantities in individuals with anemia, and is involved with the recovery of lost red blood cells. However, it is not known how the Samd14 protein plays a role in regenerating blood cells, or whether Samd14 interacts with other proteins required for red blood cell production.

To shed light on these questions, mouse cells exposed to anemia conditions were used to see what proteins Samd14 binds to. Purifying Samd14 revealed that it interacts with the actin capping protein. This interaction relies on a specific region of Samd14 that is similar to regions in other proteins that bind capping proteins. Ray et al. found that the interaction between Samd14 and the actin capping protein increased the signals needed for the development and survival of new red blood cells.

These results identify a signaling mechanism that, if disrupted, could cause anemia to develop. They lead to a better understanding of how our bodies recover from anemia, and potential avenues to treat this condition.

*Lenox et al., 2005*; *Lenox et al., 2009*; *Perry et al., 2009*). SCF is the ligand for the Kit receptor tyrosine kinase, which stimulates expansion of cultured erythroid precursors from mouse and human bone marrow via parallel MAPK and PI3K/Akt-signaling pathways (*Lennartsson and Rönnstrand, 2012*; *Munugalavadla and Kapur, 2005*). In vivo, Kit or SCF partial loss-of-function causes chronic anemia and/or acute anemia hypersensitivity (*Broudy et al., 1996*; *Geissler and Russell, 1983*; *Harrison and Russell, 1972*). Mutations that impair Kit signaling are associated with defective stress erythropoiesis and disease pathogenesis (*Agosti et al., 2009*; *Pittoni et al., 2011*). Importantly, context-dependent activation of multiple parallel pathways downstream of Kit are needed to elicit maximal Kit-mediated pro-survival and proliferation signals (*Timokhina et al., 1998*).

In a mouse model of chemical-induced acute anemia (generated by the erythrocyte lysing agent phenylhydrazine [PHZ]), Sterile Alpha Motif (SAM) domain-containing protein 14 (Samd14) expression increases more than 10-fold in erythroid precursors. Upregulation of Samd14 expression in anemia requires an intronic *cis*-regulatory element (Samd14-Enh) occupied by the transcription factors GATA2 and TAL1 (*Hewitt et al., 2017*; *Hewitt et al., 2015*). In anemia, Samd14 enhances SCF-mediated Kit signaling (*Hewitt et al., 2017*). Deletion of Samd14-Enh ($Samd14^{\Delta Enh/\Delta Enh}$) does not impact physiological hematopoiesis but causes lethality in a mouse model of severe hemolytic anemia (*Hewitt et al., 2017*). $Samd14^{\Delta Enh/\Delta Enh}$ erythroid precursors exhibit defective regenerative erythropoiesis following PHZ-induced anemia, including formation of fewer erythroid precursor (burst forming unit-erythroid [BFU-E] and colony forming unit-erythroid [CFU-E]) colonies. $Samd14^{\Delta Enh/\Delta Enh}$ mice phenocopy the cell signaling deficits and anemia sensitivity seen in the $Kit^{Y567F/Y567F}$ knock-in mice (*Agosti et al., 2009*). In both genetic models, deletion/mutation did not perturb physiological erythropoiesis (*Agosti et al., 2004*; *Hewitt et al., 2017*). Given the role of Samd14 in signaling, and similarities between Samd14- and Kit signaling-deficient mice, the mechanism in which Samd14 promotes survival is likely due in part to SCF-mediated activation of Kit signaling.

While SAMD14 mechanisms are poorly described, several emerging studies have demarcated non-malignant and malignant contexts in which deregulated SAMD14 function may be important. Human variants in the 5′ untranslated region of SAMD14, and polymorphisms attenuating SAMD14 expression, are associated with altered steady state hematologic parameters (*Astle et al., 2016*; *Fehrmann et al., 2011*). SAMD14 promoter hypermethylation and/or downregulation is associated with adenocarcinoma (*Sun et al., 2008*), gastric cancer (*Xu et al., 2020*), and the prostate tumor microenvironment (*Teng et al., 2021*). In an ex vivo genetic complementation assay to restore Samd14 expression in

$Samd14^{\Delta Enh/\Delta Enh}$ precursors, an evolutionarily conserved SAM domain in SAMD14 functions to increase Kit signal transduction and erythrocyte regeneration (*Ray et al., 2020*). Despite structural similarities, the molecular features of the SAMD14 SAM domain are distinct from SAM domains in other proteins. SAM domains have well-recognized roles in signal transduction, including cell surface receptor activation, receptor endocytosis and MAP kinase activities (*Nagamachi et al., 2013*; *Vind et al., 2020*; *Wang et al., 2018*). While Samd14 SAM domain contributes to Samd14 activities, the Samd14 protein lacking a SAM domain retains some activities, suggesting additional functional domains. Here, we discovered that SAMD14 interacts with the barbed end actin capping protein (CP) complex via a non-canonical CP-interaction motif to promote erythroid precursor activity and differentiation during stress erythropoiesis. We also provide evidence that, while Samd14 promotes both SCF- and Epo-dependent cell signaling in stress erythroid precursors, the Samd14-CP complex is required for SCF-dependent Kit signaling but not Epo signaling.

## Results

### The stress-activated Samd14 protein interacts with capping protein complex

Samd14-enhancer (Samd14–Enh) deletion in mice reduces the regenerative capacity of the erythroid system (*Hewitt et al., 2017*). In a murine Gata1-null-erythroid (G1E) cell line resembling normal proerythroblasts, we deleted the Samd14-Enh with Transcription Activator-Like Effector Nucleases (TALENs) (*Figure 1A*; *Weiss et al., 1997*). Samd14 protein was not detectable in G1E-ΔEnh cells (*Figure 1B*). To test whether Samd14 protein interacts with other proteins in proerythroblasts, we infected G1E-ΔEnh cells with GFP-tagged retroviruses expressing empty vector (EV) or hemagglutinin (HA)-Samd14. To normalize levels of exogenous protein that best mimics endogenous levels, infected cells were purified based on a $GFP^{low}$ fluorescence-activated cell sorting (FACS) gating strategy (*Figure 1—figure supplement 1A*). GFP was used as a selection marker for this and all subsequent experiments involving retroviral infection. Samd14 interacting proteins were immunoprecipitated (IPed) using anti-HA antibody-conjugated beads and enrichment scores were quantitated by mass spectrometry. We detected 21 proteins which were >twofold enriched in HA-Samd14 pull-down samples vs EV (p<0.05) (*Supplementary file 1*). Gene ontology analysis revealed enrichment of proteins involved in cell-cell adhesion (6 proteins) and barbed end filamentous (F)-actin filament capping (4 proteins). The most enriched Samd14-interacting proteins (>10-fold) (Capzβ, Capzα1, and Capzα2) belonged to the same family of F-actin CPs, which form a CP complex to regulate actin filament assembly and disassembly (*Isenberg et al., 1980*; *Schafer and Cooper, 1995*). To validate the interaction, we IPed endogenous Samd14 from wild type (WT) G1E cells using an anti-Samd14 antibody. Capzβ co-IPed with Samd14 (*Figure 1C*). To test whether the Samd14 interaction with Capzβ was SAM domain-dependent, we performed co-IPs in G1E-ΔEnh expressing either EV, HA-Samd14, or HA-Samd14 ΔSAM. No Samd14 or CP complex components were detected in IPs from EV-infected cells (*Figure 1D*). However, Capzβ, Capzα1, and Capzα2 were pulled down in both HA-Samd14- and HA-Samd14 ΔSAM-expressing cells (*Figure 1D*). These results established that Samd14 interacts with CP complex components Capzβ, Capzα1, and Capzα2 in proerythroblasts independent of the SAM domain.

### Role of capping protein complex in erythropoiesis

In muscle, CP heterodimers (Capzα and Capzβ) are localized to the Z-disk of the sarcomere where they cap barbed end actin filaments (*Casella et al., 1987*; *Solís and Russell, 2019*). CP heterodimers are components of a stable complex containing dynactin, actin-related protein (Arp1) and Arp10 (*Figure 1E*) which coordinates a wide range of cellular processes (reviewed in *Schroer, 2004*). *Capzb* expression promotes protein kinase C (PKC)-mediated signal transduction via an interaction between CP and the PKC scaffolding protein Receptor for Activated C-kinase (RACK) (*Pyle et al., 2002*). In mature erythrocytes, while CP is abundantly expressed, CP does not cap actin, but rather is localized to the cytosol unassociated with actin filaments (*Kuhlman and Fowler, 1997*). The expression, localization, and function of CP in early erythroid progenitors and precursors has not been described. To assess CP expression dynamics in erythropoiesis, we differentiated human peripheral blood CD34+ cells to erythroblasts (*Lee et al., 2015*). In unsorted cells, Capzβ protein levels decreased modestly

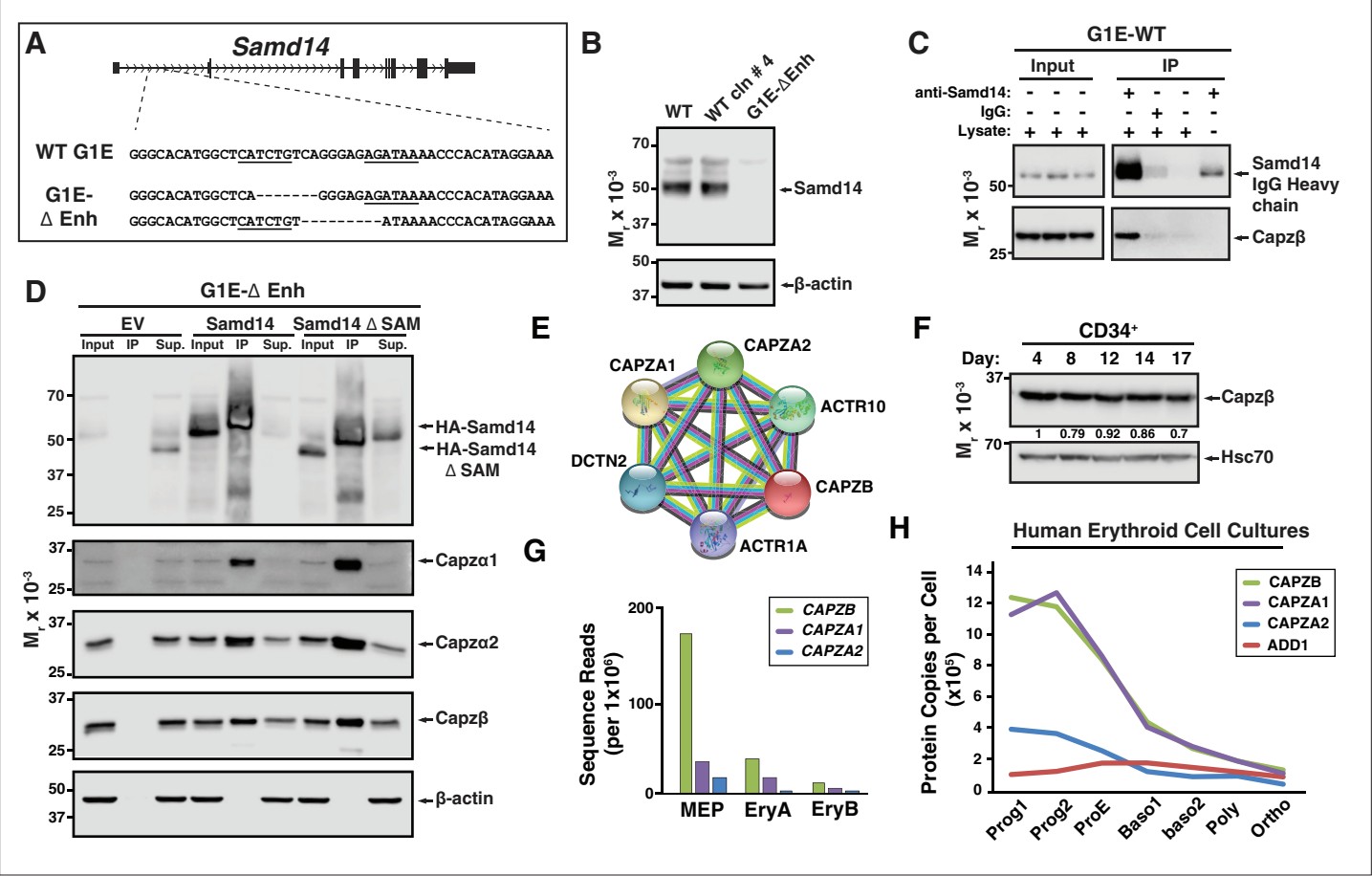

**Figure 1.** Establishing a Samd14 protein interactome in proerythroblasts. (**A**) The mouse sterile alpha motif domain 14 (*Samd14*) locus contains an E-box-GATA composite element (Samd14-Enh sequence highlighted) in intron 1 (*Hewitt et al., 2015*). The G1E wild type (WT) sequence and enhancer knockout G1E-derived cell clone sequence following TALEN directed enhancer knockout (G1E-ΔEnh) are shown. (**B**) Western blot of Samd14 and β-actin expression in G1E, WT clone #4 and G1E-ΔEnh clone. (**C**) Western blot of WT G1E cell lysates following pulldown with anti-Samd14 or anti-rabbit IgG control antibody. Input corresponds to 5% of immunoprecipitation (IP) lysate. (**D**) Western blot of G1E-ΔEnh cell lysates expressing empty vector (EV), hemagglutinin (HA)-Samd14 or HA-Samd14 Δ SAM following anti-HA pulldown. Blots were stained with anti-Samd14, anti-Capzα1, -Capzα2, -Capzβ, and β-actin antibodies. Input corresponds to 5% of IP lysate. (**E**) STRING plot depicting known interactions between capping protein (CP) complex subunits CAPZA1, CAPZA2, and CAPZB (https://string-db.org/). (**F**) Western blot and semi-quantitative densitometry analysis of human CD34+ cell lysates at 4, 8, 12, 14, and 17 days of differentiation stained with anti-Capzβ and anti-Hsc70 antibodies. (**G**) Quantitation of *Capzb*, *Capza1*, and *Capza2* mRNA transcript levels in fluorescence-activated cell sorting (FACS) purified mouse bone marrow-derived hematopoietic cells. . Data from RNA-sequencing in *Lara-Astiaso et al., 2014*. (**H**) Quantitation of CAPZB, CAPZA1, CAPZA2, and α-adducin (ADD1) protein copies per cell (*Gautier et al., 2016*). Relative levels measured by quantitative mass spectrometry to determine absolute protein levels in human erythroid progenitors throughout differentiation stages. Prog1-Band3⁻CD71medGPA⁻, Prog2- Band3⁻CD71highGPA⁻, ProE- Band3⁻CD71highGPAlow, Baso1- Band3lowCD71highGPAmed, Baso2- Band3medCD71highGPAhighCD49dhigh, Poly- Band3medCD71highGPAhighCD49dmed, and Ortho-Band3highCD71medGPAhigh. MEP: megakaryocyte erythroid progenitor; EryA: Ter119+CD71+FSChigh; EryB: Ter119+CD71+FSClow.

The online version of this article includes the following source data and figure supplement(s) for figure 1:

**Source data 1.** Source Western blot images for *Figure 1*.

**Figure supplement 1.** Expression of Samd14 Interacting proteins in hematopoietic stem/progenitor cells and precursors.

after 17 days in culture (*Figure 1F*). Mining RNA-seq data from sorted hematopoietic progenitor cells and progeny from bone marrow (*Lara-Astiaso et al., 2014*) revealed *Capzb*, *Capza1*, and *Capza2* mRNA expression in HSPCs and MEPs decrease following terminal erythroid maturation (*Figure 1G* and *Figure 1—figure supplement 1B*). Quantitative mass spectrometry in FACS-purified human erythroid cells throughout differentiation (*Gautier et al., 2016*) show a gradual decrease in the relative CP protein levels, with the concomitant increase in α-adducin (*Figure 1H*). In erythrocytes, α-adducin

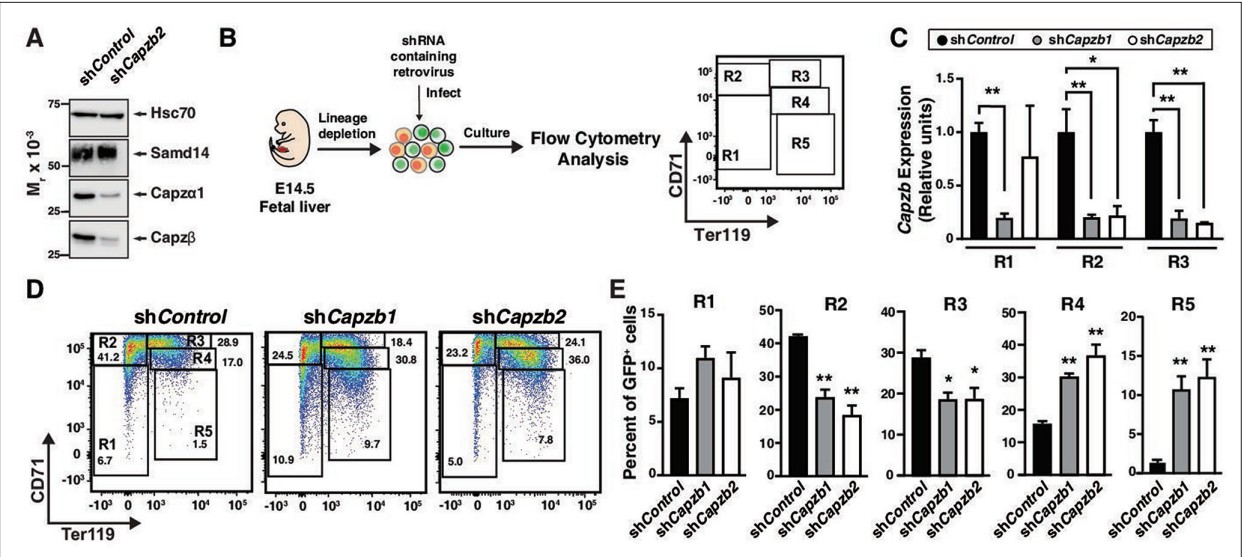

**Figure 2.** Capzβ restricts erythroid differentiation during fetal liver hematopoiesis/development. (**A**) Western blot of wild type (WT) G1E cell lysates after retroviral infection with control shRNA (sh*Control*) or shRNA targeting *Capzb* mRNA (sh*Capzb2*) stained with anti-sterile alpha motif domain 14 (Samd14), anti-Capzα1, anti-Capzβ, or anti-Hsc70 antibodies. (**B**) Experimental layout. E14.5 mouse fetal liver progenitors are retrovirally infected with sh*Control,* sh*Capzb1,* or sh*Capzb2* and cultured 3 days. R1–R5 flow cytometry gating using anti-CD71 and anti-Ter119 antibodies represents progressive stages of erythroid maturation. (**C**) Quantitation of *Capzb* mRNA in R1 (CD71$^{low}$Ter119$^{low}$), R2 (CD71$^{high}$Ter119$^{low}$), and R3 (CD71$^{high}$Ter119$^{high}$) fetal liver progenitors from WT (N=3) mice in control (sh*Control*) and following *Capzb* knockdown (sh*Capzb1* and sh*Capzb2*). (**D**) Representative flow cytometry of E14.5 fetal liver progenitors expressing shControl, sh*Capzb1,* or sh*Capzb2* using anti-CD71 and anti-Ter119 antibodies. (**E**) Quantitation of R1–R5 percentages in E14.5 fetal liver progenitors following retroviral infection with shControl, sh*Capzb1,* or sh*Capzb2* and 3-day culture (N=3). Error bars represent SD. *p<0.05; **p<0.01; ***p<0.001; ****p<0.0001 (two-tailed unpaired Student's *t* test).

The online version of this article includes the following source data for figure 2:

**Source data 1.** Source Western blot images for *Figure 2*.

(ADD1) is a known to promote spectrin-actin assembly, while CP does not appear to interact with actin filaments (*Gardner and Bennett, 1987*; *Kuhlman and Fowler, 1997*).

To investigate the role of *Capzb* in erythropoiesis, we conducted loss-of-function experiments. Knockdown of *Capzb* in G1E cells by ~80% also lowered expression of the alpha subunit to a similar degree while Samd14 expression remained unchanged (*Figure 2A*). Lower CP α1/2 expression upon loss of the β subunit is consistent with prior reports (*Sizonenko et al., 1996*). As no prior studies have tested CP function in erythropoiesis, we retrovirally infected E14.5 wildtype mice fetal liver cells with shRNA targeting *Capzb* (sh*Capzb-1 and* sh*Capzb-2*) or a control shRNA targeting the firefly luciferase gene and monitored differentiation by flow cytometry (*Figure 2B*). In CD71-expressing cells (R2 and R3), infection with either sh*Capzb-1* or sh*Capzb-2* decreased *Capzb* mRNA by ~ fivefold after 3-day culture (*Figure 2C*). To monitor cell maturation through progressive stages of erythroid maturation, 3-day expansion cultures were stained with cell surface markers CD71 and Ter119 (namely R1–R5) (*Zhang et al., 2003*) in control and sh*Capzb*-1 and sh*Capzb*-2-infected cells (*Figure 2D*). When quantitated, we observed that *Capzb* knockdown decreased the percentage of less mature R2 (CD71$^{med/high}$Ter119$^-$) cells by 2.3-fold (p=0.0013) and R3 (CD71$^{high}$Ter119$^+$) cells by 1.5-fold compared to control shRNA-infected cells. *Capzb* knockdown increased the percentages of more mature R4 (2.3-fold, p=0.0035) and R5 (8.4-fold, p=0.0086) erythroid cells (*Figure 2E*).

To test whether CP function extends to adult stress erythropoiesis, akin to Samd14, we knocked down *Capzb* in the context of acute anemia induced by PHZ. Stress erythroid progenitors and precursors were isolated from WT spleen, cultured ex vivo, and monitored for erythroid differentiation, erythroid colony forming potential and survival (*Figure 3A*; *Ray et al., 2020*). In contrast to *Samd14*, which increases upon PHZ-induced anemia, *Capzb* and *Capza2* mRNA and protein decreased after PHZ treatment in a population containing erythroid precursors (defined by surface markers CD71$^+$Kit$^+$Ter119$^-$) (*Figure 3B* and *Figure 3—figure supplement 1A and B*). *Capzb* knockdown in PHZ-treated erythroid precursors decreased *Capzb* expression twofold (p=0.016) and decreased

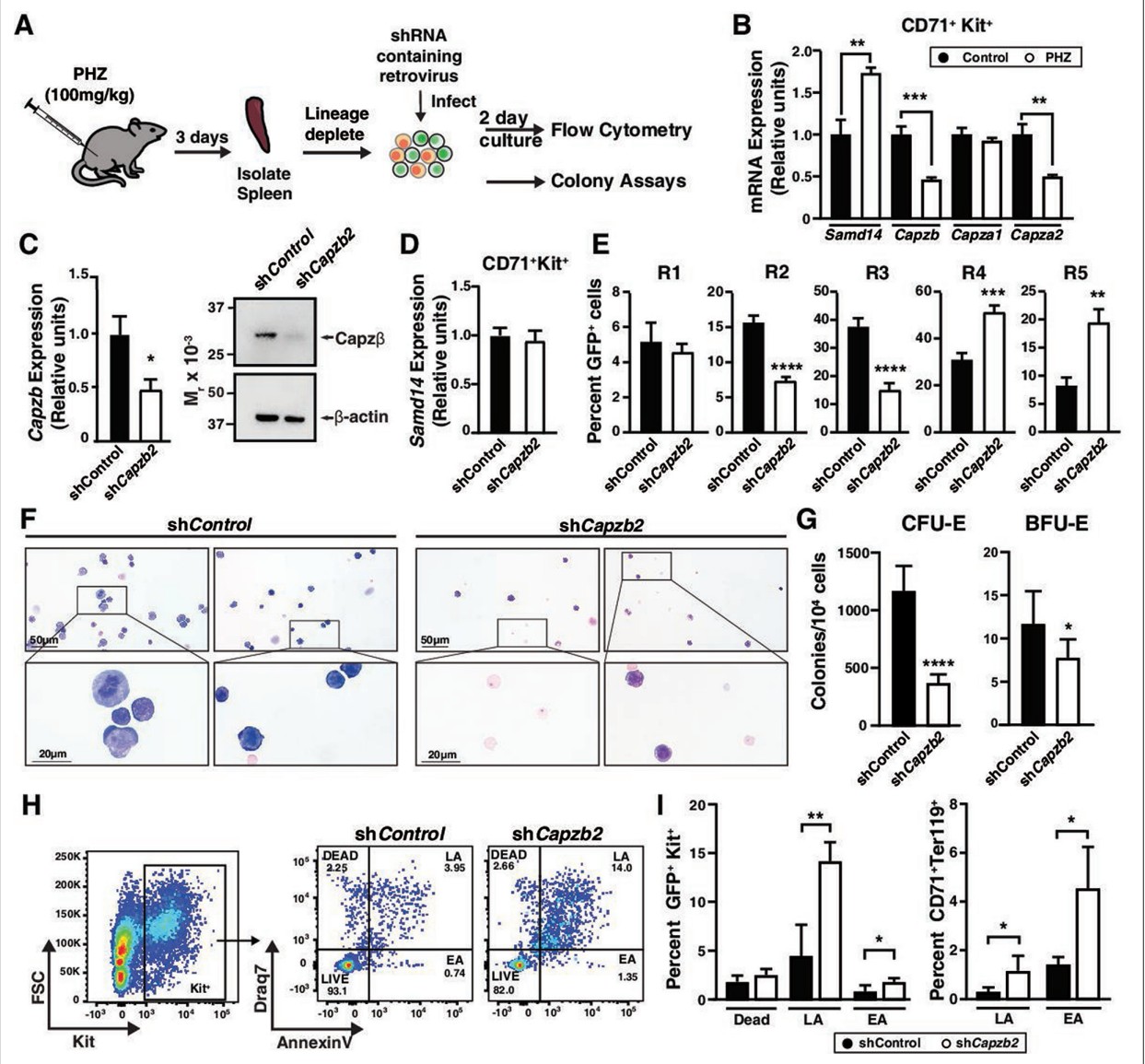

**Figure 3.** Capping protein (CP) complex promotes stress progenitor activity. (**A**) Experimental layout of ex vivo spleen retroviral infection and cultures. (**B**) Relative mRNA expression of sterile alpha motif domain 14 (*Samd14*), *Capzb*, *Capza1*, and *Capza2* in control vs phenylhydrazine (PHZ)-treated CD71⁺Kit⁺Ter119⁻ spleen cultures. (**C**) Quantitation of *Capzb* mRNA in fluorescence-activated cell sorting (FACS)-purified GFP⁺Kit⁺CD71⁺ cells from wild type (WT) mice following retroviral infection with shControl, sh*Capzb2* after 2-day culture (left); Western blot of primary spleen cultures following Capzβ knockdown (sh*Capzb2*) or sh*Control* stained with anti-Capzβ and β-actin antibodies (right). (**D**) Quantitation of *Samd14* mRNA in FACS-purified GFP⁺Kit⁺CD71⁺ cells from WT mice following retroviral infection with shControl, sh*Capzb2* after 2-day culture. (**E**) Quantitation of infected R1–R5 cells in WT PHZ-treated cells following *Capzb* knockdown. (**F**) Representative images from Wright-Giemsa stained cells following Capzb knockdown (40 × magnification)(**G**) Retrovirally-infected spleen progenitors were GFP-purified by FACS and grown for 2 days colony forming unit-erythroid (CFU-E) or 5 days burst forming unit-erythroid (BFU-E) and quantitated (N=9). (**H**) Representative flow cytometry scatter plot of membrane-impermeable DNA dye (Draq7) and anti-annexin V pacific blue (AnnV). Cells were first segregated on GFP⁺ and Kit⁺ gating. Live = Draq7⁻ AnnV⁻; Early apoptotic (EA)=Draq7⁻AnnV⁺; Late apoptotic (LA)=Draq7⁺AnnV⁺. (**I**) Quantitation of percent dead, LA and EA cells in the GFP⁺Kit⁺ cells (left) and GFP⁺CD71⁺Ter119⁺ (right). Error bars represent SD. *p<0.05; **p<0.01; ***p<0.001; ****p<0.0001 (two-tailed unpaired Student's *t* test).

The online version of this article includes the following source data and figure supplement(s) for figure 3:

**Source data 1.** Source Western blot images for *Figure 3*.

**Figure supplement 1.** CP complex function in erythropoiesis.

**Figure supplement 1—source data 1.** Source Western blot images for *Figure 3—figure supplement 1*.

Capzβ protein compared to control infections (*Figure 3C*) with no change in Samd14 expression (*Figure 3D*). *Capzb* knockdown in spleen erythroid precursors decreased the percentage of early R2/R3 by 2.1-fold and 2.44-fold and increased the percentage of R4/R5 cells by 1.65-fold and 2.35-fold compared to controls (*Figure 3E*). In *Samd14^{ΔEnh/ΔEnh}* cells, we observed similarly decreased percentages of R2/R3 and increased R4/5 cells (*Figure 3—figure supplement 1C*). Wright-Giemsa staining to assess cell morphology indicated that *Capzb* knockdown cultures contained more mature erythroblasts and reticulocytes than controls (*Figure 3F*). *Capzb* knockdown cells were also smaller overall, as measured by the decreased forward scatter-area of the cells in R2–R5 populations, consistent with increased numbers of mature cells (*Figure 3—figure supplement 1D*). These data demonstrate that *Capzb* expression opposes erythroid maturation in developing fetal liver progenitors and in splenic stress erythroid precursors. *Capzb* knockdown data are comparable to previously published data in which *Samd14* knockdown decreased R2/R3 percentages and increased R4/5 percentages in erythroid cultures (*Hewitt et al., 2015*), suggesting that Samd14 and Capzβ may have similar roles in erythropoiesis.

To determine whether CP complex regulates erythroid precursor activity, we performed colony forming unit (CFU) assays in cells infected with control or sh*Capzb2*. *Capzb* knockdown cells were plated in methylcellulose semisolid media containing SCF and Epo to promote BFU-E and CFU-E activity. Compared to controls, *Capzb* knockdown cells formed 1.5-fold fewer BFU-E (p=0.016) and threefold fewer CFU-E (*Figure 3G* and *Figure 3—figure supplement 1E*) colonies. Prior work indicated that *Capzb* depletion increased cell proliferation (*Aragona et al., 2013*). We quantitated proliferating erythroid precursor cells using Ki67 in CD71^{low}Kit^{+}. The percentage of Ki67^{+} cells were 1.6-fold higher in Capzb-depleted cells compared to controls (*Figure 3—figure supplement 1D*). To test whether *Capzb* promotes cellular viability and survival, we analyzed cultured spleen progenitors by flow cytometry for percentages of live (AnnexinV^{−}Draq7^{−}), early apoptotic (EA; AnnexinV^{+}Draq7^{−}), and late apoptotic cells (LA; AnnexinV^{+}Draq7^{+}) cells. In Capzb knockdown cells vs controls, the percentages of dead cells (Draq7^{+}) in spleen and bone marrow cultures were higher throughout immunophenotypically defined stages of erythroid maturation (*Figure 3—figure supplement 1E and F*). Correspondingly, EA (AnnexinV^{+}Draq7^{-}) and LA (AnnexinV^{+}Draq7^{+}) cells increased in *Capzb* knockdown cells vs control knockdown (*Figure 3H*). *Capzb* knockdown increased the percentage of late apoptotic cells 2.8-fold in GFP^{+}Kit^{+} and 3.68-fold in GFP^{+}CD71^{+}Ter119^{+} compared to control (*Figure 3I*). The CP complex therefore promotes erythroid precursor activity and survival following acute anemia.

## A non-canonical CP binding domain facilitates Samd14-CP complex formation and function

As a first step to establishing whether Samd14 and CP complex form a cooperative functional unit to regulate erythropoiesis in anemia, we identified candidate Samd14 sequences which may mediate capping protein binding (CPB). Since the Samd14-CP interaction occurs independent of the SAM domain (*Figure 1D*), we generated a series of Samd14 truncation mutants outside the SAM domain that corresponded with proteins known to interact with CP (*Figure 4A*). Amino acids 38–54 of Samd14 share 3 out of 7 consensus sequence residues (43%) with the CP interaction (CPI) motif LXHXTXXRPK(6×)P (X=any amino acid) (*Bruck et al., 2006*; *Edwards et al., 2015*), present in CPI proteins CP, ARP2/3 and Myosin I Linker (CARMIL), CD2-associated protein (CD2AP), Cin85, CapZ interacting protein (CapZIP), CK2-interacting protein 1 (CKIP-1; also known as PLEKHO1 (*Canton et al., 2006*). A proline-rich region of Samd14 (a.a. 114–124) aligned with five amino acids in the CPI protein Neurabin-1. Two additional proline rich regions of Samd14 aligning with CKIP-1 at residues 170–180 and 272 and 295, but not predicted to interact with CP, were deleted as controls. HA-tagged truncation constructs lacking these regions (Samd14-Δ38–54, Samd14-Δ114–124, Samd14-Δ170–180, Samd14-Δ272–295) were expressed in G1E-ΔEnh cells at similar levels (*Figure 4B*). GFP^{low} cells were FACS purified at 48 hr post-infection and Samd14-Δ38–54, Samd14-Δ114–124, Samd14-Δ170–180, and Samd14-Δ272–295 proteins were IPed using anti-HA agarose beads. When we analyzed IP proteins by Western blotting, Samd14-Δ37–55 failed to pull down Capzβ and Capzα2 in contrast to three other Samd14 deletion constructs (*Figure 4B*). Thus, amino acids 37–55 of Samd14 mediate the Samd14-CP interaction. Since this interaction domain has a low degree of sequence conformity with the canonical CPI motif, we refer to it as the Samd14 CPB domain.

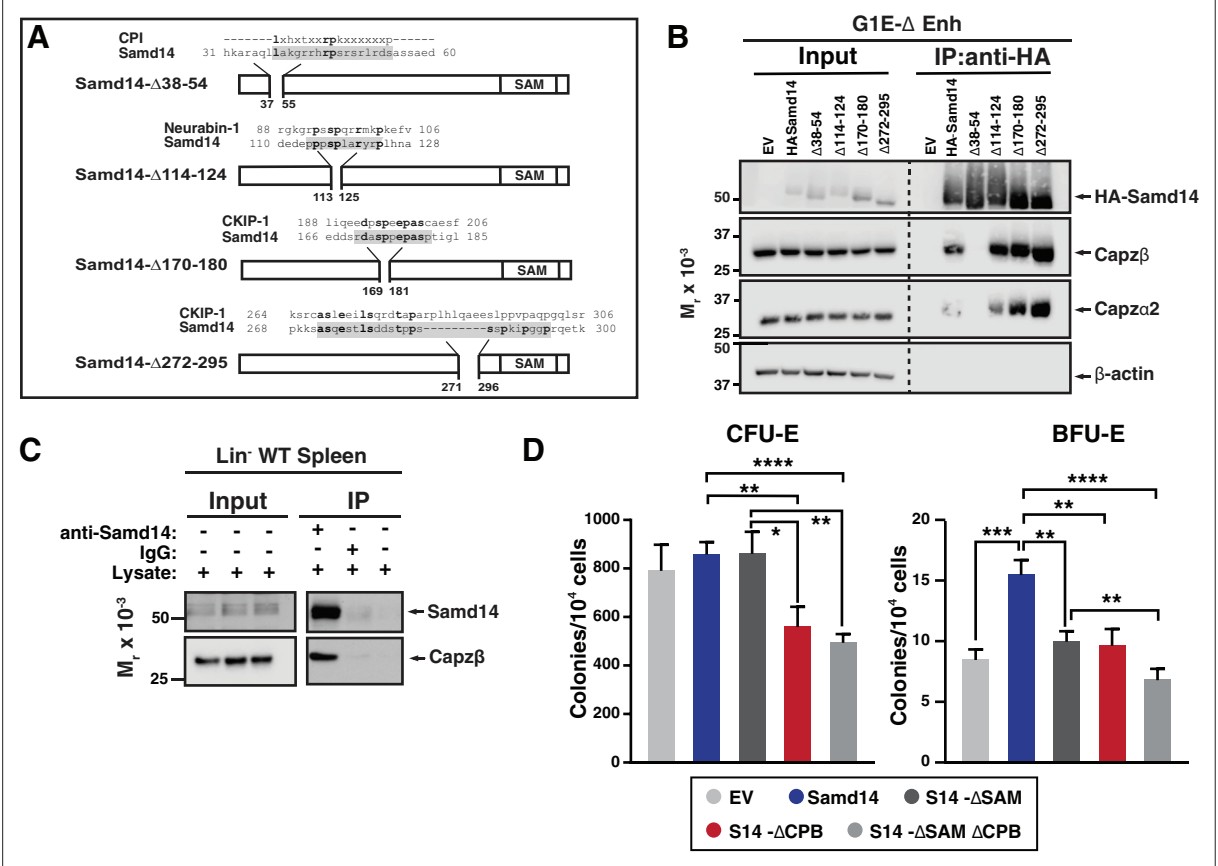

**Figure 4.** Capping protein binding (CPB) domain of sterile alpha motif domain 14 (Samd14) promotes stress progenitor activity. (**A**) Schematic representation of Samd14 deletion construct location, protein sequence, and sequence alignment with known capping protein interacting (CPI) proteins. (**B**) Western blot of G1E-ΔEnh cell lysates expressing empty vectore (EV) or hemagglutinin (HA)-tagged Samd14, Samd14- Δ38–54, Samd14-Δ114–124, Samd14-Δ170–180, or Samd14-Δ272–295 constructs immunoprecipitation (IP) with anti-HA beads and stained with anti-HA, anti-Capzβ, anti-Capzα2, anti-Capzα1, or anti-β-actin antibodies. (**C**) Western blot of Lin- WT phenylhydrazine (PHZ)-treated spleen lysates immunoprecipitated (IPed) with anti-Samd14 or anti-rabbit IgG antibodies and stained with anti-Samd14 or anti-Capzβ antibodies. (**D**) Quantitation of colony forming unit-erythroid (CFU-E) (day 2 culture) and burst forming unit-erythroid (BFU-E) (day 5 culture) colonies in GFP+ spleen progenitors expressing EV, HA-tagged Samd14, Samd14 ΔSAM, Samd14 ΔCPB, and Samd14 ΔCPBΔSAM constructs (N=6). Error bars represent SD. *p<0.05; **p<0.01; ***p<0.001; ****p<0.0001 (two-tailed unpaired Student's t test).

The online version of this article includes the following source data and figure supplement(s) for figure 4:

**Source data 1.** Source Western blot images for *Figure 4*.

**Figure supplement 1.** Dynamic regulation of Samd14-CP interaction in PHZ-induced anemia.

**Figure supplement 1—source data 1.** Source Western blot images for *Figure 4—figure supplement 1*.

In lineage-depleted spleen cells 3 days after PHZ injection, we confirmed by co-IP that the Samd14-CP interaction occurs in endogenously expressed proteins (*Figure 4C*). To determine if the Samd14-CP complex was dynamically regulated in anemia, we compared co-IP in control spleen to spleen isolated from PHZ-treated mice. In PHZ-treated spleen, more Capzb protein was pulled down compared to control spleen (*Figure 4—figure supplement 1A*), indicating that the frequency and/or stability of the Samd14-CP interaction was increased in acute anemia. No differences were seen in the amount of Capzb pulled down in CD71+Ter119- vs CD71+Ter119+ cells, or in response to acute SCF stimulation (*Figure 4—figure supplement 1B and C*). Therefore, Samd14-CP complex formation was not stage- or signal-dependent.

We previously described a SAM domain requirement for Samd14-mediated erythroid progenitor activity in PHZ-induced stress erythroid progenitors. To determine if the Samd14 CPB domain promotes unique Samd14 functions, or whether SAM and CPB domains act synergistically, we utilized an ex vivo genetic complementation assay in which Samd14 is exogenously expressed in *Samd14^ΔEnh/*

$^{\Delta Enh}$ PHZ-treated spleen erythroid progenitor cells. Primary cells infected with either full length Samd14, Samd14 lacking the SAM domain (S14-ΔSAM), CPB domain (S14-ΔCPB), or both (S14-ΔSAMΔCPB) were sorted based on GFP expression and plated in methylcellulose semisolid media for colony assays. Whereas Samd14 promoted BFU-E activity vs empty vector, S14-ΔSAM and S14-ΔCPB had fewer BFU-E colonies compared to Samd14 (1.5-fold and 1.6-fold, respectively) (*Figure 4D*). In S14-ΔSAMΔCPB-infected cells, BFU-E activity was 2.2-fold lower than Samd14 (p<0.0001). BFU-E in S14-ΔSAMΔCPB-infected cells were decreased by 1.4-fold (p=0.009) vs S14-ΔSAM, suggesting a synergistic role of both domains to promote Samd14 activity in stress BFU-E (*Figure 4D*). Consistent with prior results demonstrating that the SAM domain is not needed for CFU-E colony formation, we observed similar decreases in numbers of CFU-E in S14-ΔCPB (1.5-fold, p=0.0048) and S14-ΔSAMΔCPB (1.7-fold) expressing cells vs full length Samd14.

Given the role of Samd14 in stress-activated SCF/Kit signaling (*Hewitt et al., 2017*; *Ray et al., 2020*), we tested whether the Samd14 CPB mediates SCF-dependent activation of cell signaling in stress erythroid cultures. 48 hr after infection with either control (EV) or HA-Samd14 retrovirus, lineage-depleted splenocytes from PHZ-treated *Samd14$^{\Delta Enh/\Delta Enh}$* mice were serum-starved, stimulated with SCF, and analyzed by flow cytometry for AKT and ERK activation. The cell surface marker CD71 (transferrin receptor) can be used to distinguish among stages of erythroid differentiation (*Flygare et al., 2011*). As each cell stage has distinct cell signaling requirements, we sub-divided GFP$^+$ cells into CD71$^{low}$, CD71$^{med}$, and CD71$^{high}$ fractions, and then gated for Kit expressing cells by flow cytometry (*Figure 5A*). CD71$^{low}$ cells contained BFU-Es and very few cells expressing the mature erythroid marker Ter119$^+$, which increased in frequency in CD71$^{med}$ and CD71$^{high}$ cells (*Figure 5—figure supplement 1A*). As prior studies indicated, there is a continuum of cell phenotypes from CD71$^{low}$Kit$^+$, CD71$^{med}$Kit$^+$ to CD71$^{high}$Kit$^+$ characterized by a transition from BFU-E to CFU-E, distinct proliferative indices, and transcriptional states (*Li et al., 2019*). CD71$^{med}$Kit$^+$ cells contained more cells with CFU-E potential compared to CD71$^{high}$Kit$^+$ (*Figure 5—figure supplement 1B and C*). Samd14 expression increased the percentage of CD71$^{low}$ Kit$^+$ (1.35-fold) and CD71$^{med}$Kit$^+$ (1.5-fold) cells (*Figure 5A*). In EV-infected *Samd14$^{\Delta Enh/\Delta Enh}$*, the median fluorescence intensity (MFI) of CD71$^{low}$Kit$^+$ cells stained with phospho-ERK (pERK) or phospho-AKT (pAKT) antibodies increased by 62.7-fold and 65.3-fold, respectively, after SCF stimulation. Samd14 expression did not alter pAKT or pERK MFI in CD71$^{low}$Kit$^+$ cells (*Figure 5B and C*). Consistent with this, pERK levels were similar between EV-infected WT and *Samd14$^{\Delta Enh/\Delta Enh}$* cells, indicating that Kit signaling in the CD71$^{low}$ population is insensitive to Samd14 expression (*Figure 5C*). In CD71$^{med}$ cells, pERK levels were 4.3-fold (p<0.0001) lower in EV-infected *Samd14$^{\Delta Enh/\Delta Enh}$* cells compared to WT EV-infected cells, indicating that CD71$^{med}$ cells are sensitive to Samd14 expression levels (*Figure 5D*). pERK MFI in *Samd14$^{\Delta Enh/\Delta Enh}$* CD71$^{med}$ cells was rescued by Samd14 (*Figure 5B and D*). Samd14 expression increased the levels of pERK (4.4-fold) and pAKT (15.3-fold) in SCF-treated CD71$^{med}$ cells compared to EV controls (*Figure 5D*). In CD71$^{high}$ cells, pAKT and pERK levels in Kit$^+$ cells do not change after SCF stimulation. However, pAKT and pERK levels in CD71$^{high}$Kit$^+$ cells are sensitive to Samd14 expression (*Figure 5E*). These results indicated that distinct Kit$^+$ populations gated on CD71 show varying signaling response (as measured by pAKT and pERK) to SCF and Samd14, but do not distinguish whether Kit$^+$ cells in the CD71$^+$ gates are completely unresponsive to SCF stimulation or whether Kit signaling is occurring via other pathways. Finally, we tested whether the SAM or CPB domains of Samd14 could alter Samd14 sensitivity in the SCF-responsive CD71$^{med}$Kit$^+$ cells. Deleting either the SAM or the CPB domain of Samd14 reduced levels of pAKT (fourfold) and pERK (twofold) compared to Samd14 in the CD71$^{med}$ population (*Figure 5F*). As an alternative assessment of Kit signal-promoting roles of the Samd14 CPB domain, we examined the activation of Kit transcription in response to SCF stimulation (*Figure 5—figure supplement 1D*). Whereas Kit transcript levels were significantly increased in Samd14-expressing cells vs controls, we did not detect significant increases in Kit transcript levels in S14-ΔCBP expressing cells (*Figure 5—figure supplement 1D*). Both the SAM and CPB domains of Samd14 are required for maximal promotion of Kit signaling in stress erythroid precursors.

## Structural and functional requirements of the Samd14 CPB domain

The Samd14 CPB sequence, which is required for SCF/Kit signaling and Samd14-dependent erythroid precursor activities, is conserved in amphibians, reptiles, birds, and mammals (*Figure 6A*). When we aligned functionally-validated CPI domains identified in CPI proteins Twinfilin (Twf1) (*Johnston*

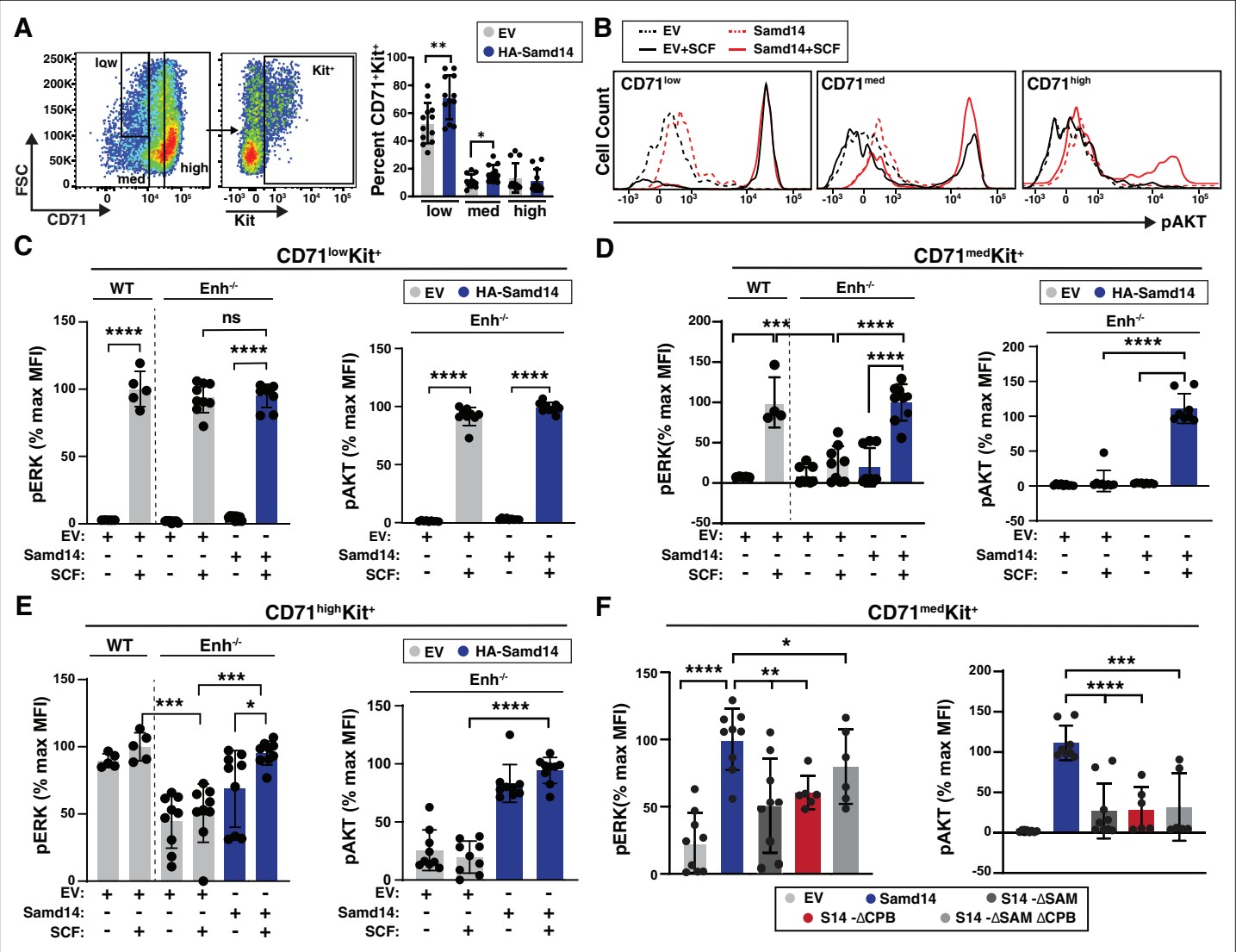

**Figure 5.** Sterile alpha motif domain 14 (Samd14)-capping protein (CP) enhances stem cell factor (SCF)/Kit signaling in CD71med stress progenitors. (A) Flow cytometry gating to analyze retrovirally infected (GFP+) cells based CD71low, CD71med and CD71high, and Kit+ (left); Percent GFP+CD71+Kit+ cells in CD71low, CD71med, and CD71high fractions in phenylhydrazine (PHZ)-treated spleens expressing empty vector (EV) or hemagglutinin (HA)-Samd14 (N=9) (right). (B) Histograms depicting pAKT fluorescence in GFP+CD71lowKit+, GFP+CD71medKit+, and GFP+CD71highKit+ erythroid precursors in unstimulated (dotted line) or post-SCF stimulation (5 min, 10 ng/ml) (solid line). (C) Quantitation of pERK1/2 (left) and pAKT (right) median fluorescence intensity (MFI) in GFP+CD71lowKit+ cells from wild type (WT) or Samd14- $Samd14^{\Delta Enh/\Delta Enh}$ spleen expressing EV or HA-Samd14 at 5 min post-SCF stimulation. (D) Quantitation of pERK1/2 (left) and pAKT (right) MFI in GFP+CD71medKit+ cells from WT or $Samd14^{\Delta Enh/\Delta Enh}$ spleen expressing EV or HA-Samd14 at 5 min post-SCF stimulation. (E) Quantitation of pERK1/2 (left) and pAKT (right) MFI in GFP+CD71highKit+ cells from WT or $Samd14^{\Delta Enh/\Delta Enh}$ spleen expressing EV or HA-Samd14 at 5 min post-SCF stimulation. (F) Quantitation of pERK1/2 and pAKT MFI in GFP+CD71medKit+ retrovirally infected spleen cells with EV, HA-tagged Samd14, Samd14 ΔSAM, Samd14 ΔCPB and Samd14 ΔCPBΔSAM (N=6). Error bars represent SD. *p<0.05; **p<0.01; ***p<0.001; ****p<0.0001 (two-tailed unpaired Student's *t* test).

The online version of this article includes the following figure supplement(s) for figure 5:

**Figure supplement 1.** Linking progenitor activity and transcriptional activation in immunophenotypically defined populations.

et al., 2018), CapZIP, WASHCAP (*Hernandez-Valladares et al., 2010*), CKIP-1 (*Canton et al., 2006*), CARMIL1 (*Yang et al., 2005*), CIN85, and CD2AP (*Bruck et al., 2006*), each contain a central arginine (R) or lysine (K) containing R/KXR/K motif (*Figure 6B*). Given the low sequence identity between Samd14 CPB and the other identified CPI motifs, we conducted a proteome-wide search for similar peptide sequences. The Samd14 CPB aligned to a region of Neurabin-1 (64.7% sequence identity) (*Figure 6B*). Interestingly, Neurabin-1 also contains a SAM domain. In functional studies of CPI domains,

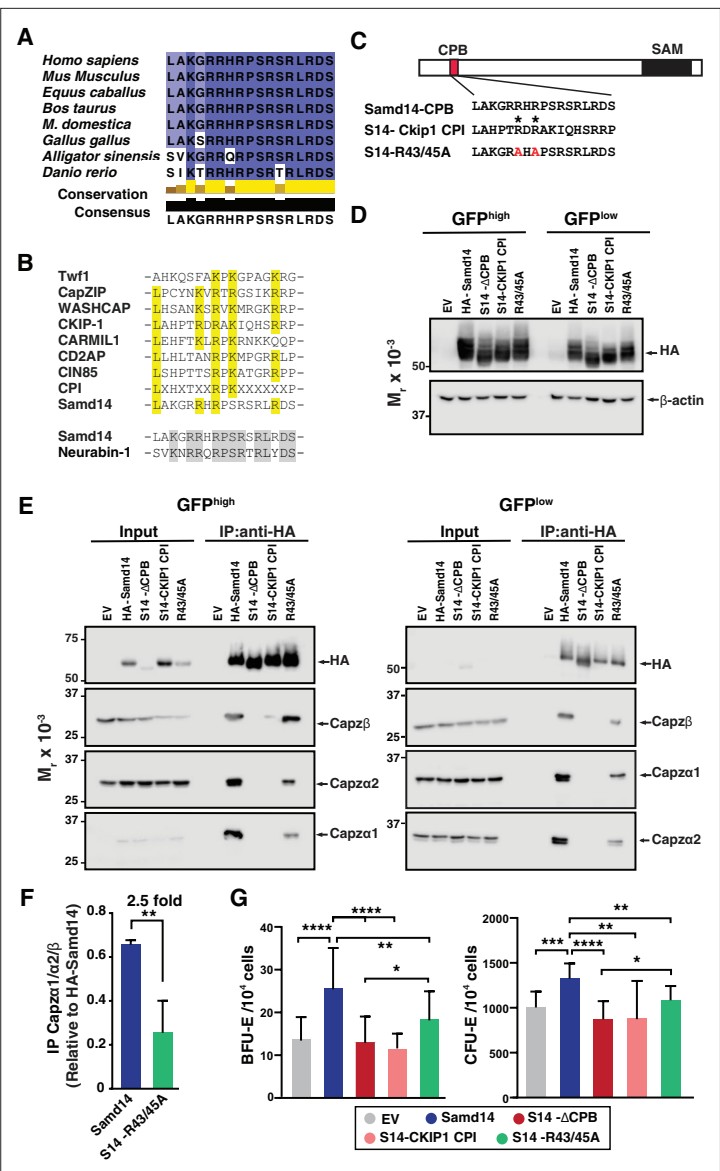

**Figure 6.** Atypical capping protein interaction (CPI) domain of sterile alpha motif domain 14 (Samd14) conferring stress erythroid precursor activity. (**A**) Sequence alignment of the capping protein binding (CPB) domain of Samd14 among vertebrate species. Aligned using Clustal Omega and visualized with JalView. Yellow bars in conservation plot represents evolutionary conservation. (**B**) Sequence alignment of the capping protein interaction (CPI) motif in other proteins with Samd14 CPB domain. Yellow color indicates conserved residues among the different CPI motifs. (**C**) Schematic representation of chimeric and mutated CPB domain Samd14 constructs. (**D**) Western blot of GFP-sorted G1E-ΔEnh cells expressing empty vector (EV), hemagglutinin (HA)-Samd14, Samd14-ΔCPB, Samd14(S14)-Ckip1-CPI, or Samd14-R43/45 A stained with anti-HA. HA and β-actin (**E**) Western blot analysis of Capzα1, Capzα2, and Capzβ co-immunoprecipitated (IPed) following pull-down of EV, HA-tagged Samd14, Samd14-ΔCPB, Samd14(S14)-Ckip1-CPI and Samd14-R43/45 A with anti-HA agarose beads from retrovirally infected G1E-ΔEnh cells. (**F**) Densitometry analysis of the relative CapZ co-IPed with either HA-Samd14 or HA-S14-R43/45 R. (**G**) Quantitation of GFP⁺ colony forming unit-erythroid (CFU-E) (N=9) and burst forming unit-erythroid (BFU-E) (N=17) colonies formed in spleen progenitors retrovirally infected with EV, HA-tagged Samd14, Samd14-ΔCPB, Samd14(S14)-Ckip1-CPI, and Samd14-R43/45 A. Error bars represent SD. *p<0.05; **p<0.01; ***p<0.001; ****p<0.0001 (two-tailed unpaired Student's *t* test).

The online version of this article includes the following source data for figure 6:

**Source data 1.** Source Western blot images for *Figure 6*.

central arginine residues are required to mediate CP complex interaction (*Johnston et al., 2018*). To determine if the aligned arginine residues in the Samd14 CPB domain mediate CP complex binding we mutated the R43/R45 to alanine (Samd14-R43/45 A) in an expression construct (*Figure 6C*). To test whether the Samd14-CPB domain function could be replaced by known CPI domains in other proteins, we substituted 17-amino acids in the Samd14-CPB domain with another known CPI motif in CKIP-1 (23.5% protein sequence identity, maintaining central arginine residues). We performed an IP Western blot of HA-tagged Samd14-R43/45 A and the Samd14-Ckip1 CPI proteins expressed in G1E-ΔEnh cells. GFP^high and GFP^low FACS sorted cells were prepared and analyzed separately to assess whether high/low expression levels of exogenous protein may alter binding affinity (*Figure 6D*). Surprisingly, the Samd14-Ckip1 CPI chimeric protein failed to pull down any CP complex subunits (Capzβ, Capzα1, and Capzα2) (*Figure 6E*). Whereas Samd14-R43/45 A still interacts with the CP complex, the mutation reduced the amount of CP complex components pulled down (*Figure 6F*). Thus, Samd14 R43/45 residues mediate CPB affinity but are not required. Next, we tested whether expression of Samd14-Ckip1 CPI or Samd14-R43/45 A (which interfere with Samd14 binding to CP) impairs Samd14 function. PHZ-treated *Samd14^{ΔEnh/ΔEnh}* spleen progenitors expressing Samd14-Ckip1 CPI formed 2.23-fold and

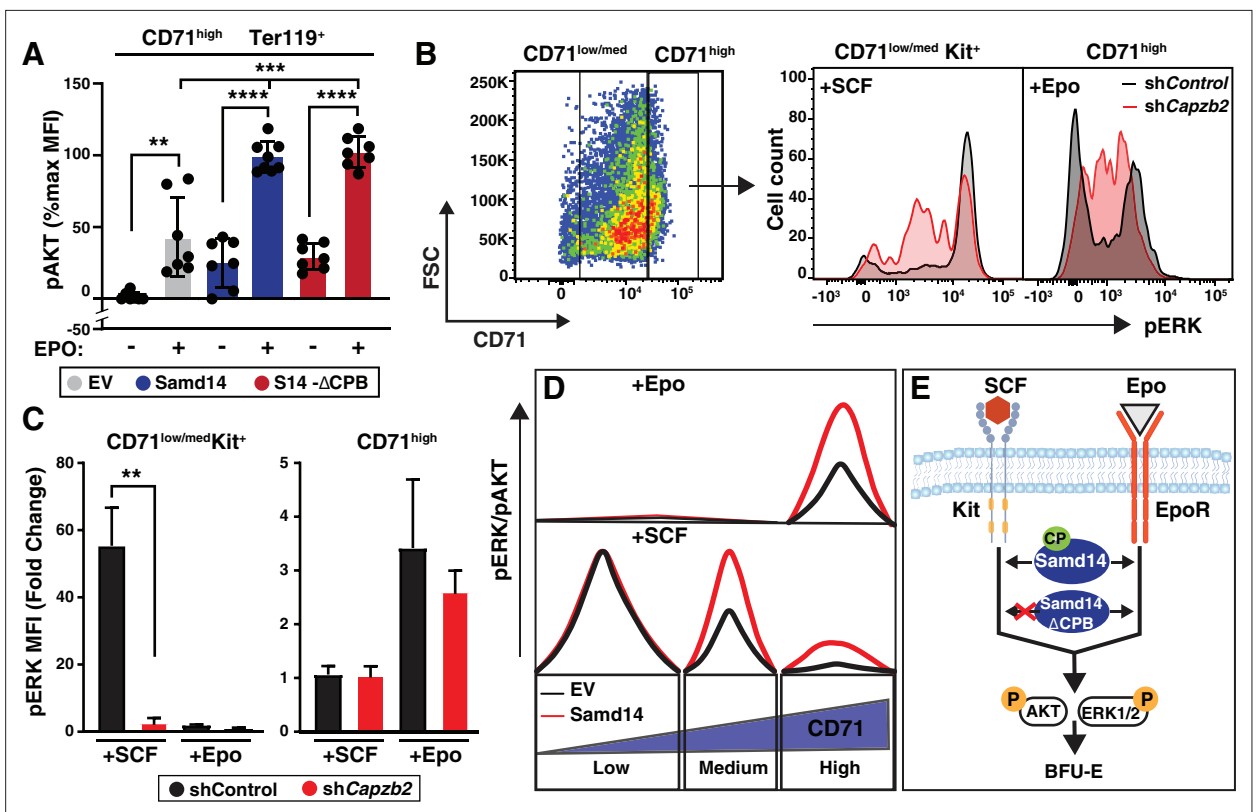

**Figure 7.** Sterile alpha motif domain 14 (Samd14) increases erythropoietin (Epo) signaling independent of capping protein (CP) complex interaction. (**A**) Quantitation of pAKT MFI in GFP⁺CD71^highTer119⁺ retrovirally infected *Samd14^{ΔEnh/ΔEnh}* spleen cells expressing empty vector (EV), hemagglutinin (HA)-Samd14 or Samd14-ΔCPB at 10 min post-Epo (5 U/ml) stimulation (N=7). (**B**) Flow cytometry gating to analyze infected (GFP⁺) phenylhydrazine (PHZ)-treated cells expressing sh*Control* or sh*Capzb2* based on CD71^low/med and CD71^high; representative histograms depict pERK1/2 fluorescence in GFP⁺CD71^low/medKit⁺ cells after stem cell factor (SCF) stimulation (10 ng/ml, 5 min) and GFP⁺CD71^high cells after Epo stimulation (10 min, 5 U/ml). (**C**) Quantitation of pERK1/2 MFI after SCF (5 min, 10 ng/ml) or Epo (10 min, 5 U/ml) stimulation in GFP⁺CD71⁺Kit⁺ and GFP⁺CD71^high spleen cells expressing sh*Control* or sh*Capzb2* (N=3). (**D**) CD71^low cells are SCF-responsive but Samd14-insensitive, CD71^med are SCF-responsive and Samd14-sensitive, and CD71^high cells are unresponsive to SCF. CD71^low/med cells responding to SCF do not respond to Epo stimulation, whereas the CD71^high cells are Epo dependent and Samd14-sensitive. (**E**) Samd14-CP interaction promotes burst forming unit-erythroid (BFU-E) colony formation, SCF/Kit and Epo/EpoR signaling by phosphorylating ERK1/2 and AKT in PHZ-treated spleen cells, in the absence of the Samd14-CP interaction PHZ-treated spleen cells fails to promote SCF/Kit signaling, but Epo signaling remains unaffected. Error bars represent SD. *p<0.05; **p<0.01; ***p<0.001; ****p<0.0001 (two-tailed unpaired Student's *t* test).

The online version of this article includes the following figure supplement(s) for figure 7:

**Figure supplement 1.** Samd14-CP enhances SCF/Kit signaling.

1.5-fold fewer BFU-E and CFU-E than Samd14, respectively (*Figure 6G*). However, cells expressing Samd14-Ckip1 CPI and Samd14-ΔCPB formed similar numbers of colonies (*Figure 6G*). The Samd14-R43/45A-expressing cells (with reduced but not absent CPB) formed fewer BFU-E and CFU-E colonies compared to Samd14 but more than Samd14-ΔCPB (*Figure 6G*). These results demonstrate the importance of the Samd14-CP interaction for Samd14-mediated stress erythroid precursor activity. Compared to other functionally validated CPB domains, the Samd14 CPB domain has additional sequence attributes that enable the Samd14-CP protein interaction.

Given that Samd14 promoted SCF signaling within a narrow window of CD71 expression, it is possible that Samd14 regulates other signaling pathways at distinct stages of erythropoiesis. Erythropoietin signaling regulates erythroid maturation and cell survival during stress as well as normal erythropoiesis (*Liu et al., 2006*; *Xiang et al., 2015*). To test whether Samd14 regulates Epo signaling, we serum starved and Epo stimulated cultures expressing either EV, Samd14, or Samd14-ΔCPB and assessed AKT activation. pAKT increased 2.28-fold (p=0.0002) in Samd14-expressing cells vs EV control (*Figure 7A*). Similarly, pERK increased 5.85-fold in Samd14 expressing cells vs EV controls (*Figure 7—figure supplement 1*). Surprisingly the Samd14-ΔCPB also increased the pAKT levels by 2.3-fold (p=0.0002). This contrasts with our results in SCF stimulated cells, suggesting that the Samd14-CP interaction drives Samd14-dependent Kit signaling but is dispensable for Epo signaling (*Figure 7A*). To test the involvement of *Capzb* in cell signaling, we analyzed ERK1/2 activation levels upon SCF/Epo stimulation after *Capzb* knockdown. SCF responsive cells (CD71$^{low}$, CD71$^{med}$, *Figure 5B*) were analyzed together as GFP$^+$CD71$^{low/med}$Kit$^+$ cells (*Figure 7B*). *Capzb* knockdown decreased SCF-dependent pERK1/2 activation by 19.3-fold (p=0.001) relative to sh*Control* (*Figure 7C*) and reduced the overall magnitude (MFI) of SCF-mediated pERK1/2 activation (*Figure 7—figure supplement 1B*). Following Epo stimulation, the CD71$^{high}$ fraction (which is unresponsive to SCF) increased ERK1/2 phosphorylation (*Figure 7C*). Epo-stimulated pERK1/2 was unaffected by *Capzb* knockdown in CD71$^{high}$ cells. A similar trend was seen using pAKT as a readout for Epo signaling (*Figure 7—figure supplement 1C*). These data define cell stage-specific requirements for Samd14 in both SCF/Kit and Epo/EpoR signaling pathways (*Figure 7D*). Whereas *Capzb* expression and the CPB domain of Samd14 promote Kit signaling, the interaction between Samd14 and CP is not an important mediator of Epo signaling (*Figure 7E*).

## Discussion

To reveal mechanisms of anemia stress-activated protein Samd14, and its function in cell signaling, we analyzed Samd14-interacting proteins. We report that Samd14 interacted with a family of F-actin CPs (Capzβ, Capzα1, and Capzα2) in a SAM domain-independent manner. These subunits form the heterodimeric actin CP complex that binds the barbed end of actin during filament assembly/disassembly and plays a role in cell morphology, migration, membrane trafficking, and signaling (*Canton et al., 2006*; *Terry et al., 2018*). Mature erythrocytes contain an $\alpha_1\beta_2$ conformation of CP that is present in the cytosol and not associated with the short actin filaments in the erythrocyte membrane skeleton (*Kuhlman and Fowler, 1997*). While pointed end actin CPs like Tropomodulin (Tmod3) coordinate critical steps in late-stage erythropoiesis (*Sui et al., 2014*), the role of barbed end capping molecule CP in erythropoiesis was not previously investigated. We show here that *Capzb* expression opposed erythroid differentiation in fetal liver and stress erythroid precursors, regulated cell size, and promoted cell survival and precursor colony forming activity. The Samd14-CP interaction occurs through a CPI-like domain to mediate Samd14-dependent functions in signaling and progenitor activities. While our studies implicate CP as a mediator of erythropoiesis, the exact mechanism is unclear. Among the potential functions of CP, it is a core component of the dynactin complex which controls early endosome dynamics (*Valetti et al., 1999*). This represents one potential mechanism whereby CP may function in erythroid precursor cell signaling.

The rate of erythropoiesis can increase by 15–20-fold under stress conditions. Numerous proteins and factors are involved in mechanisms which accelerate, maintain, and ultimately resolve stress erythropoiesis in anemia (*Bresnick et al., 2018*; *Paulson et al., 2020*). Stress erythroid progenitors and precursors in spleen and liver have unique cellular behaviors and respond to distinct paracrine signals relative to bone marrow-derived erythroid progenitors (*Harandi et al., 2010*). Illustrating the requirements for Kit signaling in anemia, SCF deletion in spleen endothelial cells attenuates the anemia stress response and reduces the red blood cell counts without affecting bone marrow hematopoiesis (*Inra*

*et al., 2015*). Additional stress-specific signaling molecules BMP-4, hedgehog glucocorticoid and GDF15 promote stress erythroid progenitor expansion and homing (*Bauer et al., 1999*; *Hao et al., 2019*; *Lenox et al., 2005*; *Perry et al., 2009*). Transcriptional mechanisms are also intimately involved in erythroid progenitor/precursor stress responses. The transcription factor GATA1 promotes stress erythropoiesis in addition to roles in developmental and physiological erythropoiesis (*Gutiérrez et al., 2008*). GATA1 and related transcription factor GATA2 directly activate Samd14 transcription through an intronic enhancer which is present in a network of other genes predicted to control vital aspects of stress erythropoiesis (*Hewitt et al., 2017*; *Hewitt et al., 2015*). GATA2 and Samd14, as well as anti-apoptotic genes such as *Bcl2l1*, are transcriptionally elevated in stress erythropoiesis via similar E-box-GATA composite element sequences (*Hewitt et al., 2017*). As one node of this network, our results in a stress progenitor/precursor genetic rescue system have provided rigorous evidence for a mechanism in which Samd14 upregulation drives stress progenitor activity through its CPI to coordinate stress-dependent signaling mechanisms.

As a facilitator of cell signaling that is upregulated in response to anemia, Samd14 represents a new constituent driving anemia-specific Kit activities. These conclusions are typified in the $Samd14^{\Delta Enh/\Delta Enh}$ mouse, in which the E-box-GATA composite element is selectively required for stress progenitor responses/activities and anemia-dependent Kit signaling. Similar decreases in stress progenitor activities were observed in other genetic models which attenuate Kit activation (e.g. the $Kit^{Y567F}$ mouse) (*Agosti et al., 2009*). To establish whether Samd14 mechanisms integrate in other pathways, we also describe a role for Samd14 in Epo signaling. Whereas *Capzb* knockdown lowered Epo signaling, there were no differences in the ability of full length Samd14 and Samd14-ΔCPB to activate Epo signaling. Thus, both Samd14 and CP appear to have complex-independent functions in erythroid precursor activity and cell signaling. Together, our findings suggest new models for regulation of cell signal transduction in stress erythropoiesis. We envision two important functions, not necessarily exclusive, that result from the interaction of Samd14 with CP through its CPB domain. First, CP may target Samd14, and therefore, specific signaling activities, at specific locations in the cell important for function, such as membrane-associated receptor complexes. Second, the Samd14-CP interaction may influence activities of the larger CP-containing dynactin complex in endocytic trafficking, impacting the rate of ligand-activated receptor turnover in erythroid cell membranes. It is attractive to propose that the temporary upregulation of Samd14 in acute anemia may be tied to a burst of erythropoiesis by controlling the rate of receptor turnover. The model system used in these studies examines stress-activated erythroid precursors after they have expanded. While this permitted the ex vivo analysis of stress BFU-E in culture systems, fewer BFU-E and CFU-E suggests that Samd14-CP mechanisms may play a role in earlier immature progenitor cells to oppose erythroid maturation. Importantly, while we used PHZ injection to induce acute anemia, *Samd14* expression is also elevated in clinically relevant anemia models induced by severe bleeding and erythroid radioprotection, suggesting that Samd14 mechanisms are not specific to this model. These experiments have taken important steps toward elucidating Samd14 function in acute anemia, the mechanism whereby the Samd14-CP interaction controls erythroid precursor cell signaling, and how cell signaling networks are optimally coordinated in stress erythropoiesis to accelerate and resolve acute anemia.

## Materials and methods

**Key resources table**

| Reagent type (species) or resource | Designation | Source or reference | Identifiers | Additional information |
|---|---|---|---|---|
| Gene (*Mus musculus*) | Samd14 | GenBank | Gene ID: NM_146025.2 | |
| Genetic reagent (*Mus musculus*) | Wild type (C57BL/6 J) | Jackson Laboratory | Strain #:000664 | |
| Genetic reagent (*Mus musculus*) | Samd14-Enh⁻/⁻ | PMID:28787589 | | |
| Cell line (*Mus msculus*) | G1E | PMID:9032291 | | Gift from M. Weiss |

*Continued on next page*

*Continued*

| Reagent type (species) or resource | Designation | Source or reference | Identifiers | Additional information |
|---|---|---|---|---|
| Antibody | anti-Samd14 (rabbit polyclonal) | PMID:28787589 | Bresnick lab | WB (1:2000) IP (5 µg) |
| Antibody | anti-HA (rabbit monoclonal) | Cell Signaling Technology | Cat.: #3724; RRID:AB_1549585 | WB (1:2000) |
| Antibody | anti-capzα1 (rabbit polyclonal) | ThermoFisher | Cat.: #PA5-31026 RRID: AB_2548500 | WB (1:1000) |
| Antibody | anti-capzα2 (rabbit polyclonal) | ThermoFisher | Cat.: PA5-29982; RRID: AB_2547456 | WB (1:1000) |
| Antibody | anti-capzβ (mouse monoclonal) | Santa Cruz Biotechnology | Cat.: #sc-136502; RRID:AB_10610091 | WB (1:1000) |
| Antibody | anti-capzβ (rabbit polyclonal) | Bethyl Lab | Cat.: #A304-734A-M | WB (1:1000) |
| Antibody | Mouse anti-Rabbit IgG (Light-Chain Specific) | Cell Signaling Technology | Cat.: #93702; RRID:AB_2800208 | WB (1:500) |
| Antibody | anti-phospho (S473)-AKT (p-AKT) (rabbit monoclonal) | Cell Signaling Technology | Cat.: #4060; RRID:AB_2315049 | FC (1:100) |
| Antibody | anti-phospho (Thr202/Tyr204) p44/42 ERK1/2 (p-ERK) (rabbit monoclonal) | Cell Signaling Technology | Cat.: #9101; RRID:AB_331646 | FC (1:100) |
| Antibody | APC-goat anti rabbit IgG | Jackson ImmunoResearch | Cat.: #111-136-144; RRID:AB_2337987 | FC (1:200) |
| Antibody | PE/Cyanine7(PECy7) anti mouse CD117 (c-Kit) (rat monoclonal) | Biolegend | Cat.: #105,814 | FC (1:200) |
| Antibody | APC anti-mouse Ter119 (rat monoclonal) | Biolegend | Cat.: #116,211 | FC (1:200) |
| Antibody | PE-anti mouse CD71 (rat monoclonal) | Biolegend | Cat.: #113,807 | FC (1:200) |
| Commercial assay/kit | MojoSort Streptavidin Nanobeads | Biolegend | Cat.: #480,016 | FC (1:200) |
| Antibody | APC anti-mouse Ki-67 (rat monoclonal) | Biolegend | Cat.: #652,406 | FC (1:200) |
| Peptide, recombinant protein | Pacific Blue Annexin V | Biolegend | Cat: #640,917 | FC (1:50) |
| Antibody | anti-rabbit IgG isotype | Invitrogen | Cat: #10,500 C | IP (5 µg) |
| Recombinant DNA reagent | pMSCV PIG (plasmid) | Addgene | Plasmid #21654; RRID:Addgene_21654 | |
| Peptide, recombinant protein | Recombinant mouse SCF protein | R&D Systems | Cat.: #455-MC-050 | |
| Chemical compound, drug | Phenylhydrazine | Sigma Aldrich | P26252 | |
| Software, algorithm | Flowjo 10.6.2 | https://www.flowjo.com/solutions/flowjo. | RRID:SCR_008520 | |
| Software, algorithm | Fiji | https://imagej.net/software/fiji/ | | |
| Software, algorithm | GraphPad Prism | https://www.graphpad.com/ | RRID:SCR_002798 | |

SCF: stem cell factor; Samd14-Enh$^{-/-}$: sterile alpha motif domain 14-enhancer.

## Plasmids and Samd14 constructs

HA-tagged full-length Samd14, Samd14 lacking the SAM domain (Samd14 ΔSAM) (*Ray et al., 2020*), Samd14 lacking the CPB domain (Samd14 ΔCPB), Samd14 lacking both SAM and CPB domains (Samd14 ΔCPBΔSAM) were cloned into mammalian expression plasmid pMSCV-Puro-IRES-GFP (PIG) (Addgene #21654) using BglII and EcoRI restriction digest sites. Other Samd14 truncation mutants (Samd14-Δ38–54, Samd14-Δ114–124, Samd14-Δ170–180, Samd14-Δ272–295) were similarly cloned into pMSCV PIG. HA-tagged chimeric protein in which coding sequences corresponding to 17 amino acid CPI domain of CKIP-1 (from amino acid residues 150–166) replaced the 17 amino acid Samd14 CPB domain at the exact location, termed S14-Ckip CPI, was synthesized (Twist Bioscience) and cloned

into pMSCV PIG using BglII and EcoRI restriction digest sites. All plasmids were retrovirally packaged with pCL-Eco (Addgene #12371) in 293T cells. For *Capzb* knockdown, two shRNA targeting exon 5 and exon 6 were synthesized along with a control shRNA targeting the luciferase gene as a 97-nt ultramer (Sigma) containing 22-mer reverted repeats, a 19-nt spacer and restriction sites BglII/XhoI and cloned into pMSCV PIG. shCapzb1: Exon 5 22_mer: TGGAGTGATCCTCATAAAGAAA. shCapzb2: Exon 6 22_mer: CGGTGATGCTATGGCTGCAAAC.

## Mice and primary cell isolation

Samd14-Enh mutant (*Samd14$^{\Delta Enh/\Delta Enh}$*) mice were generated by site-directed TALEN deletion (*Hewitt et al., 2017*). All animal experiments were carried out with ethical approval of the Institutional Animal Care and Use Committee at the University of Nebraska Medical Center. Fetal liver was isolated at E14.5 from WT mice. Hemolytic anemia was induced by a single dose of freshly prepared PHZ (100 mg/kg) (Sigma) administered subcutaneously in sterile PBS. Spleens were harvested from 8- to 12-week-old mice 3 days post-PHZ injection. Spleens from WT or *Samd14$^{\Delta Enh/\Delta Enh}$* (8–13 weeks old) mice were dissociated, resuspended in PBS with 2% fetal bovine serum (FBS) and passed through a 35 µm nylon filter to obtain single-cell suspensions. Erythroid precursors were isolated using negative selection by lineage-depletion with biotin-conjugated antibodies and MojoSort streptavidin-conjugated magnetic nanobeads (Biolegend): anti-mouse CD3e (clone 145–2 C11), anti-mouse CD11b (clone M1/70), anti-mouse CD19 (clone 6D5), anti-mouse CD45R (B220) (clone RA3-6B2), anti-mouse Gr-1 (clone RB6-C5), anti-mouse Ter119. Following depletion, erythroid precursors were cultured in StemPro-34 media (Invitrogen) containing 2 mM L-glutamine, Pen-Strep, 0.1 mM monothioglycerol, 1 µM dexamethasone, 0.5 U/ml erythropoietin, and 1% mSCF Chinese Hamster Ovary cell conditioned medium. For retroviral infections, $1 \times 10^6$ lineage- cells were added to 100 µl viral supernatant, polybrene (8 µg/ml) and HEPES buffer (10 µl/ml), and spinoculated at 2600 rpm for 90 min at 30°C. Cells were maintained at a density between 0.5 and $1\times10^6$ cells/ml for 2–3 days.

## G1E cell culture

*Gata1*-null, mouse embryonic stem cell-derived G1E cells (gift from the Mitch Weiss lab) resemble normal proerythroblasts (*Weiss et al., 1997*). Cells were maintained in IMDM supplemented with 15% fetal bovine serum, 0.5% SCF-conditioned media from CHO cells, 0.1 mM monothioglycerol (Sigma) and 2 U/ml erythropoietin (Amgen). TALEN-mediated deletion of Samd14-Enh was conducted as previously described (*Hewitt et al., 2015*). Retroviral infections were performed by spinoculation ($1200 \times g$ for 90 min at 30°C) of $1 \times 10^6$ G1E cells with 100 µl viral supernatant, polybrene (8 µg/ml) and HEPES buffer (10 µg/ml). FACS-purification of GFP$^+$ cells was performed 48 hr after infection on a FACSArialI.

## CD34$^+$ cell differentiation

Peripheral blood CD34$^+$ cells (Fred Hutchinson) were cultured using a four-phase differentiation protocol as described by *Lee et al., 2015*. Briefly, an initial expansion phase (day 0–4) was followed by three successive differentiation phases, differentiation I (day 5–9), differentiation II (day 10–13), and differentitation III (day 13–21). The basal media contained IMDM, 15% FBS, 2 mM glutamine, holo transferrin (500 µg/ml), and recombinant human insulin (10 µg/ml). The media was supplemented with: 1 µM β-estradiol, 5 ng/ml rhIL-3, 100 ng/ml rhSCF, and 6 U/ml rhEpo for differentiation I; with 50 ng/ml rhSCF and 6 U/ml rhEpo for differentiation II; and with 2 U/ml rhEpo for differentitation III. Cell numbers were maintained at 0.5–$1\times10^6$ cells/ml.

## Immunoprecipitation

For endogenous Samd14 immunoprecipitation, 1–$2 \times 10^7$ cells were lysed for 1 hr in 1% NP-40 Lysis buffer (150 nM NaCl, 50 mM Tris pH 8.0, 2 mM DTT, 0.2 mM PMSF, 20 µg/ml leupeptin) on ice. Cell lysates were incubated with 5 µg of either anti-Samd14 antibody (*Hewitt et al., 2017*) or anti-rabbit IgG isotype (Invitrogen, #10500 C) control overnight. Protein A beads (ThermoFisher #15918–014) were blocked for 1 hr in 1%BSA/PBS prior to IP. Cell lysates were incubated with 60 µl of the preblocked Protein A beads for 2 hr at 4°C. Beads were washed with 1% NP-40 Lysis buffer, boiled in SDS lysis buffer and analyzed by SDS-PAGE.

For HA pulldown experiments in G1E, $5 \times 10^7$ GFP$^+$ G1E-ΔEnh cells expressing different Samd14 mutants were collected between days 12 and 14 after sorting for GFP using FACS. Cells were lysed 1 hr in 1% NP-40 Lysis buffer (150 nM NaCl, 50 mM Tris pH 8.0, 2 mM DTT, 0.2 mM PMSF, 20 µg/ml leupeptin) on ice. Protein lysates were incubated with 40 µl slurry of HA-agarose beads (Pierce) overnight. Washed beads were boiled in SDS lysis buffer and analyzed by SDS-PAGE or mass spectrometry.

## Mass spectrometry

In-gel samples were reduced with 10 mM DTT and then alkylated with 10 mM iodoacetamide, washed and then digested overnight using trypsin. Tryptic digests were run by LC-MS/MS using a 2 hr gradient on a 0.075 mm × 250 mm Waters CSH C18 column on a U3000 nanoRSLC (Dionex) coupled to a Q-Exactive HF (ThermoScientific) mass spectrometer. Peptides were identified using Mascot (Matrix Science; v.2.6.1), searched with a fragment ion mass tolerance of 0.060 Da and a parent ion tolerance of 10.0 PPM. Deamidated asparagine and glutamine, oxidation of methionine and carbamidomethyl of cysteine, phospho serine, threonine and tyrosine were specified in Mascot as variable modifications. Results were then validated and summarized into Scaffold (Proteome Software; v. 4.8.9) using a 99.0% protein probability with a minimum of two unique peptides with at least 80% peptide probability. Statistically enriched proteins were determined using a 2-way ANOVA.

## Live cell flow cytometry

Retrovirally-infected cells were stained with antibodies specific for surface proteins like PE–CD71 (R17217), APC- Ter119 (116212), PE-Cy7- Kit (clone 2B8) (105814), Pacific Blue Annexin V (640917) antibodies for 30 min at 4°C (all from Biolegend). Cells were resuspended with either DAPI or Draq7 (Biolegend) viability dyes as required and analyzed with BD LSR Fortessa. FACS for GFP$^+$ cells or CD71- and Ter119-fractionation was conducted on a FACSAria II (BD Life Sciences). Data was analyzed with FlowJo v10.6.2.

## Western blotting

Proteins were boiled in SDS buffer (25 mM Tris, pH 6.8, 2% β-mercaptoethanol, 3% SDS, 5% bromophenol blue, 5% glycerol) for 10 min, resolved by SDS-PAGE, and detected with Pierce ECL Western blotting substrate (Thermo Scientific) on a LICOR imager. Primary antibodies: polyclonal anti-Samd14 (*Hewitt et al., 2017*), anti-HA (Cell Signaling Technology), anti-β-actin (Cell Signaling Technology), anti-capzα1 (Proteintech and ThermoFisher PA5-31026), anti-capzα2 (ThermoFisher, PA5-29982) anti-capzβ (Santa Cruz; Bethyl Laboratories, A304), anti-Hsc70 (Santa Cruz). Secondary antibodies goat-anti-mouse-IgG-HRP, goat-anti-rabbit-IgG-HRP (Jackson Labs), mouse anti-rabbit IgG (Light-Chain Specific) (#93702; Cell Signaling Technology).

## Quantitative PCR

Total RNA was purified from 0.5 to $2 \times 10^6$ cells in 1 ml TRIzol (ThermoFisher). For cDNA synthesis 1 µg of RNA was incubated with 200 ng oligo(dT) and 50 ng random hexamer primers at 68°C for 10 min. RNA was incubated with M-MLV reverse transcriptase (NEB) with 10 mM dithiothreitol (DTT), RNAsin (Promega), and 0.5 mM deoxynucleoside triphosphates at 42°C for 1 hr, and then heat inactivated at 98°C for 5 min. Real time PCR was performed with 2 µl cDNA, Power SYBR Green (ThermoFisher) and 200 nM of the appropriate primers on the Quant Studio 3 real time PCR system (Applied Biosystems). Relative expression level for each gene was quantified based on a standard curve of serially diluted control cDNA. Negative control reactions were performed on samples in the absence of reverse transcriptase.

## Colony formation assays

Spleen cells were isolated from mice 3 days after subcutaneous 100 mg/kg PHZ injection. Lineage-depleted, retrovirally-infected spleen progenitors were cultured for 16–20 hr prior to FACS-purification of GFP + cells. $2 \times 10^4$ GFP$^+$ cells/ml were mixed with Methocult M3434 (STEMCELL Technologies) containing Epo, SCF, IL-3 and IL-6 and plated in replicate 35 mm dishes ($2 \times 10^4$ cells in 1.0 ml) or 6-well plates ($1 \times 10^4$ cells in 0.5 ml). BFU-E colonies were counted 5 days after culturing. CFU-E colonies were counted 2 days after plating.

## Phospho-flow cytometry

After 2-day culture, primary spleen cells were serum-starved for 2 hr in 1% BSA/IMDM at 37°C and treated with 10 ng/ml SCF or vehicle for the indicated time. For Epo stimulation (5 U/ml), cells were positively selected by Biotin-Ter119 and MojoSort streptavidin-conjugated magnetic beads. Cells were immediately fixed in 2% paraformaldehyde for 10 min at 37°C and permeabilized in 95% methanol overnight at –20°C. Cells were stained with rabbit antibodies against phospho (S473)-AKT (p-AKT) and phospho (Thr202/Tyr204) p44/42 ERK1/2 (p-ERK) (4060, 9101; Cell Signaling) for 30 min, then incubated in APC-conjugated goat anti-rabbit (1:200), PE-Cy7-conjugated Kit (1:200) and PE-conjugated CD71 (1:200) for 30 min at room temperature. Samples were analyzed using a BD LSR II flow cytometer and gates were selected based on fluorescence minus one control. Values for pAKT and pERK levels were calculated by MFI using FlowJo v10.6.2 (BD Life Sciences) and normalized to the maximum overall value within each experiment (Relative MFI) or as fold change of activation over vehicle treated controls.

## Statistical analysis

Statistical analysis was carried out by GraphPad Prism v8 as indicated in the figure legends and represented by $*p<0.05$; $**p<0.01$; $***p<0.001$; $****p<0.0001$.

## Acknowledgements

This study was supported by grants to KJH (NIH/National Heart, Lung, and Blood Institute R01 HL155439-01, National Institute of Diabetes and Digestive and Kidney Diseases K01DK113117-03), funding as a project leader in the Nebraska Center for Molecular Target Discovery and Development (DHHS/NIH/NIGMS, 1P20GM121316-01-A1), and funding from the State of Nebraska-Nebraska Stem Cell Research (LB606). The Proteomics & Metabolomics Facility (RRID:SCR_021314), Nebraska Center for Biotechnology at the University of Nebraska-Lincoln is supported by the Nebraska Research Initiative. We also thank the UNMC Flow Cytometry Research Facility, supported by the Nebraska Research Initiative (NRI) and The Fred and Pamela Buffett Cancer Center's National Cancer Institute Cancer Support Grant. We thank Dr. Keith R Johnson for thoughtful critiques on this manuscript.

## Additional information

### Funding

| Funder | Grant reference number | Author |
| --- | --- | --- |
| NHLBI Division of Intramural Research | R01 HL155439-01 | Suhita Ray<br>Linda Chee<br>Yichao Zhou<br>Meg A Schaefer<br>Kyle J Hewitt |
| GMS | 1P20GM121316-01-A1 | Kyle J Hewitt |
| Nebraska Stem Cell Research | LB606 | Suhita Ray<br>Linda Chee<br>Yichao Zhou<br>Meg A Schaefer<br>Michael J Naldrett<br>Sophie Alvarez<br>Nicholas T Woods<br>Kyle J Hewitt |
| National Institute of Diabetes and Digestive and Kidney Diseases | K01DK113117-03 | Kyle J Hewitt |

The funders had no role in study design, data collection and interpretation, or the decision to submit the work for publication.

## Author contributions
Suhita Ray, Conceptualization, Data curation, Formal analysis, Investigation, Methodology, Writing - original draft, Writing – review and editing; Linda Chee, Investigation, Methodology; Yichao Zhou, Meg A Schaefer, Methodology; Michael J Naldrett, Data curation, Formal analysis, Methodology; Sophie Alvarez, Data curation, Formal analysis; Nicholas T Woods, Conceptualization, Formal analysis, Methodology; Kyle J Hewitt, Conceptualization, Formal analysis, Investigation, Methodology, Resources, Supervision, Writing – review and editing

## Author ORCIDs
Suhita Ray http://orcid.org/0000-0003-0887-6640
Kyle J Hewitt http://orcid.org/0000-0003-1946-625X

## Ethics
This study was performed in accordance with the recommendations in the Guide for the Care and Use of Laboratory Animals of the National Institutes of Health. All of the animals were handled according to approved institutional animal care and use committee (IACUC) protocols (#18-099-08 FC) of the University of Nebraska Medical Center.

## Decision letter and Author response
Decision letter https://doi.org/10.7554/eLife.76497.sa1
Author response https://doi.org/10.7554/eLife.76497.sa2

# Additional files

## Supplementary files
• Transparent reporting form

• Supplementary file 1. Samd14-interacting proteins. Mass spectrometry analysis of IP protein in EV and Samd14 conditions (N=3). Spectral count of each protein, as determined by Scaffold. Statistical significance between EV and HA-Samd14 conditions determined by Scaffold.

## Data availability
The mass spectrometry proteomics data was deposited to the ProteomeXchange Consortium via the PRIDE (Perez-Riverol et al, 2019) partner repository with the dataset identifier PXD030467 and 10.6019/PXD030467. All other data generated or analysed during this study are included in the manuscript and supporting file.

The following dataset was generated:

| Author(s) | Year | Dataset title | Dataset URL | Database and Identifier |
|---|---|---|---|---|
| Perez-Riverol et al | 2019 | The mass spectrometry proteomics data was deposited to the ProteomeXchange Consortium | http://proteomecentral.proteomexchange.org/cgi/GetDataset?ID=PXD030467 | PRIDE, 10.6019/PXD030467 |

The following previously published dataset was used:

| Author(s) | Year | Dataset title | Dataset URL | Database and Identifier |
|---|---|---|---|---|
| Lara-Astiaso D, Weiner A, Lorenzo-Vivas E, Zaretsky I, Jaitin DA, David E, Keren-Shaul H, Mildner A, Winter D, Jung S, Friedman N, Amit I | 2014 | Chromatin state dynamics during blood formation | https://www.ncbi.nlm.nih.gov/geo/query/acc.cgi?acc=GSE60103 | NCBI Gene Expression Omnibus, GSE60103 |

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
