## [Editor Report]

This study describes a new mechanism involving Samd14 and F-actin capping protein (CP) complex formation in the regulation of stress erythropoiesis. Through orthogonal biochemical, cellular, and genetic complementation assays, the authors provide evidence that the Samd14-CP interaction is required for the proper regulation of SCF/Kit signaling in erythroid precursors in response to acute anemia. The Samd14-CP-mediated mechanism may have implications in modulating the receptor tyrosine kinase signaling in physiological and pathological conditions.

---

## [Decision Letter]

**Decision letter after peer review:**

Thank you for submitting your article "Functional Requirements for a Samd14-Capping Protein Complex in Stress Erythropoiesis" for consideration by *eLife*. Your article has been reviewed by 3 peer reviewers, including Jian Xu as the Reviewing Editor and Reviewer #1, and the evaluation has been overseen by Utpal Banerjee as the Senior Editor. The following individual involved in review of your submission has agreed to reveal their identity: Robert Paulson (Reviewer #3).

The reviewers have discussed their reviews with one another, and the Reviewing Editor has drafted this to help you prepare a revised submission. Besides the essential revisions listed below, we also provide the comments from individual reviewers for your reference as you prepare the revisions.

Essential revisions:

1. Additional data are needed to support the role of Capzb in erythropoiesis. Suggested studies include cytospins, the viability and proliferation of the Capzb KD cells, and/or overexpression studies. It would be of interest to also examine some of the effects caused by Capzb knockdown in Samd14-Enhancer knockout cells.

2. Obtain additional insights into the mechanisms that contribute to the erythroid stage-specific effects. For example, Samd14 is significantly upregulated during stress erythropoiesis but CP expression is not affected. Is the Samd14-CP interaction via CPB dynamically regulated by stress erythropoiesis and/or by Kit signaling in a differentiation stage-specific manner?

3. Functional assays (colony forming ability, proliferation, etc) or alternate assessment of Kit/EPO signaling would help further support conclusions.

4. Apart from pAKT and pERK, it would be helpful to obtain additional mechanistic insights into how the Samd14-CP complex regulates Kit signaling. The authors should consider one or few of the suggested experiments such as the use of Kit expression cell line and the effects on gene expression with knockdown of the capping proteins,

5. Given the observation that BFU-Es are down, it would be helpful to determine whether the early Kit+Sca1+CD133+ progenitors are affected, for example the expansion of these cells in culture with knockdown of the capping complex.

*Reviewer #1 (Recommendations for the authors):*

This is a well-developed manuscript with a number of interesting findings. The functional and biochemical studies of Samd14-CP interaction in erythropoiesis in response to stress-induced signaling were well executed, and the results were appropriately discussed. This reviewer has a few points to further enhance the strengths of the main conclusions.

1. The authors provided evidence that Capza/b knockdown largely phenocopies Samd14 deficiency in PHZ-induced erythroid stress. To provide additional evidence for the Samd14-CP complex interaction in stress erythropoiesis, it would be of interest to perform Capza/b knockdown in Samd14-Enhancer knockout cells. These studies may further reveal whether the CP complex regulates erythroid cell maturation in a Samd14-dependent manner.

2. The model that Samd14-CP interaction regulates c-Kit but not Epo signaling is based mostly on genetic complementation assay with expression of WT or various Samd14 mutants, as well as the downstream effects on signaling cascades (i.e. phosphorylation of AKT and ERK1/2). While informative, it would be interesting to obtain further insights into the mechanisms that may contribute to the stage-specific effects. For example, Samd14 is significantly upregulated during stress erythropoiesis but CP expression is not affected. Is the Samd14-CP interaction via CPB dynamically regulated by stress erythropoiesis and/or by c-Kit signaling in a differentiation stage-specific manner?

3. Given that the main findings were based on PHZ-mediated anemic stress, it would be helpful for the authors to further discuss their findings in the context of more physiologically relevant models of stress erythropoiesis and pathologic settings.

*Reviewer #2 (Recommendations for the authors):*

A few more granular comments for the authors:

It is not clear from the apoptosis analyses discussed in 238-246 if there was an overall increase in cell death, or only in early/ late apoptosis. Looking at the specific viability of the R2/R3 cells would help to understand if they are maturing at an increased rate, or if they are dying. Also, it's not clear from the data presented that the role of Capzb is limited to maintaining viability in context of anemic stress. Does KD of Capzb in non-stressed erythroblasts or cell lines (like G1E or CD34)+ lead to loss of viability?

For the studies in Figure 5A, it is not clear what cell populations are being studied. Only a fairly narrow window of erythroblasts are typically positive for both Cd71 and kit (CFU-E and ProE), so it seems somewhat counterintuitive to parse this population further by CD71 level.

shRNA can have off target effects in erythroid cells. Use of alternate methods such as TALENs or CRISPR/Cas9 to disrupt Capzb would further support the conclusions that the CP complex has a role in erythropoiesis.

It is interesting that the CP components are downregulated following anemic stress, while SAMD14 goes up. How do these expression changes fit into the model of SAMD14-CP interaction in stress erythroid progenitors? Did this change occur on the protein level as well?

*Reviewer #3 (Recommendations for the authors):*

The paper from Ray et al., does an excellent job of identifying new components of the Kit/SCF signaling pathway. The authors have built on previous data showing the Samd14 plays a key role in regulating increased activation of Erk and Akt signaling pathways downstream of Kit receptor during stress erythropoiesis. The CP complex was not previously identified as a component of this pathway. The biochemical analysis is a strength. The weaknesses of the paper have more to do with the phenotypic analysis. The paper does not extend the Samd14 findings that Erk and Akt signaling is affected. It would be greatly improved, if there were mechanistic data showing how the Samd14-CP complex interaction affected Kit signaling. The second weakness is in the analysis of stress erythropoiesis. The data from Samd14 and the knockdown of Capz show a defect in BFU-E. However, the analysis focuses on later stage cells and on cells isolated from spleens after the expansion of the stress BFU-E population has occurred. Furthermore, stress BFU-E respond to different conditions (Epo + BMP4 + SCF at 1% O2) than steady state BFU-E (Epo + IL-3 or SCF), the conditions used here were for steady state BFU-E so some aspects of the role of the CP complex in stress erythropoiesis may be lost. The discussion would also be improved if the role of Samd14/CP complex was put into the developmental context of stress erythropoiesis. The fact that you have fewer BFU-E and when you look at erythroblasts in your cultures, they skew towards more mature cells suggests the defect is in the proliferation of immature cells and the ability of these cells to resist differentiation.

---

## [Author Response]

Essential revisions:1. Additional data are needed to support the role of Capzb in erythropoiesis. Suggested studies include cytospins, the viability and proliferation of the Capzb KD cells, and/or overexpression studies. It would be of interest to also examine some of the effects caused by Capzb knockdown in Samd14-Enhancer knockout cells.

Since the role of Capzb in erythropoiesis had not been studied, we conducted additional experiments to confirm that Capzb knockdown alters erythroid maturation in erythroid precursors. The revised manuscript extends our analysis in the following ways:

1) Cytospins and Wright-Giemsa staining were conducted on control and shCapzb knockdown cells to compare cellular morphology (Figure 3F). Capzb knockdown cultures contained more cells consistent with the morphology and staining of mature erythroblasts and reticulocytes than controls, supporting flow cytometry data that Capzb knockdown cells are smaller and more differentiated than control cells.

2) We quantified cell viability in control vs shCapzb cells. At all immunophenotypically-defined stages of erythroid differentiation, we observed an increased accumulation of dead (DAPI+) cells in shCapzb knockdown vs control. These findings correlated with increased late apoptosis in Figure 3.

3) We assessed proliferation using a Ki67 stain in shCapzb vs control cells. In CD71low/med cells, proliferation was higher in Capzb knockdown cells. These results are consistent with prior reports in adherent cell cultures, in which knockdown increased proliferation (Aragona et al., Cell 2013).

4) We tested whether Capzb knockdown elicited similar or distinct responses in Samd14-Enh^-/-^ cells as wildtype cells. Like WT cells, Capzb knockdown in Enh^-/-^ spleen erythroid precursors decreased R2/3 cell percentages and increased R4/R5 cell percentages compared to controls. Capzb knockdown in Enh^-/-^ cells decreased CFU-E potential vs. WT controls. These results suggested that the role of Capzb in erythropoiesis extends beyond just the Samd14-CP complex interaction. Experiments are quantitated in Supplemental Figure 2 and discussed in the manuscript on page x, line x. Given the wide-ranging role of CP in cell biology, and CP complex function in diverse cell activities, these results are not surprising.

2. Obtain additional insights into the mechanisms that contribute to the erythroid stage-specific effects. For example, Samd14 is significantly upregulated during stress erythropoiesis but CP expression is not affected. Is the Samd14-CP interaction via CPB dynamically regulated by stress erythropoiesis and/or by Kit signaling in a differentiation stage-specific manner?

We tested stress-specific, stage-specific and signal activation-dependent changes to the interaction between Samd14 and Capzb by endogenous Co-immunoprecipitation. Comparing WT Lin- spleen cells from control vs. PHZ-treated mice, we detected a 1.7-fold increase in Capzb pulled down. These results indicate that either the frequency or the affinity of the Samd14-CP interaction increased in anemia. To test whether the Samd14-CP interaction changes in distinct stages of erythropoiesis, we sorted CD71+Ter119- cells and CD71+Ter119+ cells and conduced IP analyses. In both R2 (CD71med/+Ter119-) and R3 (CD71+Ter119+) cells, Samd14 and Capzb were immunoprecipitated at similar ratios, indicating that the Samd14-CP interaction is not regulated in a stage-specific manner. To test the potential contribution of Kit pathway activation on the Samd14-CP interaction, Lin- cells were serum starved for 1 hour and then stimulated with SCF (10ng/mL) for 5 min. We compared the relative amount of protein immunoprecipitated by semiquantitative densitometry (N=3) which indicated there was no apparent change in the Samd14-CP interaction affinity after pathway activation. Collectively, these results suggest that the affinity of the Samd14-CP interaction increased in stress erythropoiesis but is not dynamically-regulated by erythropoietic stage or activated by SCF stimulation.

3. Functional assays (colony forming ability, proliferation, etc) or alternate assessment of Kit/EPO signaling would help further support conclusions.

To independently assess Kit activation, we tested SCF/Kit signaling induction of *Kit* primary transcript RNA. We previously noted that SCF stimulation increases *Kit* transcript levels in control cells but not Samd14 knockdown cells (Hewitt et al., Mol Cell 2015). Serum starved cells expressing EV, Samd14 or S14-ΔCPB were stimulated with 10 ng/mL SCF for 1 hour. In Enh-/- cells expressing full length Samd14, *Kit* primary transcript levels increased modestly (1.4-fold) but significantly (p=0.02), whereas Kit primary transcript levels did not increase in control empty vector-infected cells or cells expressing S14-ΔCPB. This provides an alternative read-out that demonstrates the requirement of the CPB domain of Samd14 to promote Kit signaling in stress erythroid progenitors.

4. Apart from pAKT and pERK, it would be helpful to obtain additional mechanistic insights into how the Samd14-CP complex regulates Kit signaling. The authors should consider one or few of the suggested experiments such as the use of Kit expression cell line and the effects on gene expression with knockdown of the capping proteins,

This revision incorporates several of the suggested reviewer experiments which were not included in the original manuscript. We tested whether Capzb function was specific to stress erythroid progenitors, or may apply more broadly in hematopoiesis, by shRNA mediated knockdown of Capzb in primary isolations of mouse WT bone marrow. This data showed that, while Samd14 expression is required for stress erythropoiesis but dispensable for steady-state erythropoiesis, the role of Capzb is not limited to anemic stress. Knockdown impaired cell viability and increased differentiation in bone marrow as well as spleen erythroid progenitors. As suggested by one reviewer, we conducted functional colony formation assays on immunophenotypically-defined cell populations (Kit^+^CD71^low/med/high^) to correlate the unique signaling requirements of these cells with a progenitor phenotype. Populations were characterized by increasing fraction of Ter119+ cells dependent on CD71 levels. The highest number of CFU-E were in the CD71^med^Kit^+^ population, a population that was both SCF-responsive and Samd14 sensitive in phospho-flow cytometry assays. CD71high cells, which were responsive to Epo but not SCF, contained the highest number of Ter119+ cells but fewer CFU-E.

5. Given the observation that BFU-Es are down, it would be helpful to determine whether the early Kit+Sca1+CD133+ progenitors are affected, for example the expansion of these cells in culture with knockdown of the capping complex.

To determine whether the numbers of early stress erythroid progenitors correspond to changes in BFU-E activity, we performed multiple flow cytometric analyses (with biological repeats) of splenic cells at different time points (days 0, 1, 2, and 3) in our ex vivo system. We were able to detect Kit+Sca1+CD133+ progenitors in control bone marrow and spleen. However, following PHZ treatment, uncultured and cultured cells from spleen exhibited considerable levels of autofluorescence (especially in the Sca1 channel) which obscured any potentially measurable differences in these cell numbers among experimental conditions. This observation had been previously observed in our hands and noted in Bennet et al., (*Methods Mol Biol* 2019). Importantly, all flow cytometry analyses in this study use single stain and fluorescence minus one (FMO) controls to set gates and we do not detect significant autofluorescence after staining with antibodies targeting other surface markers (CD71, Ter119, Kit) or intracellular proteins (Ki67, pAKT, pERK) used to gate out various erythroid populations in other experiments.

Reviewer #1 (Recommendations for the authors):1. The authors provided evidence that Capza/b knockdown largely phenocopies Samd14 deficiency in PHZ-induced erythroid stress. To provide additional evidence for the Samd14-CP complex interaction in stress erythropoiesis, it would be of interest to perform Capza/b knockdown in Samd14-Enhancer knockout cells. These studies may further reveal whether the CP complex regulates erythroid cell maturation in a Samd14-dependent manner.

As described in “essential revisions” we conducted these experiments and data is shown in Supplemental Figure 2

2. The model that Samd14-CP interaction regulates c-Kit but not Epo signaling is based mostly on genetic complementation assay with expression of WT or various Samd14 mutants, as well as the downstream effects on signaling cascades (i.e. phosphorylation of AKT and ERK1/2). While informative, it would be interesting to obtain further insights into the mechanisms that may contribute to the stage-specific effects. For example, Samd14 is significantly upregulated during stress erythropoiesis but CP expression is not affected. Is the Samd14-CP interaction via CPB dynamically regulated by stress erythropoiesis and/or by c-Kit signaling in a differentiation stage-specific manner?

As discussed in “essential revisions”, we conducted additional experiments to test stage-dependent, signal-dependent and stress-dependent changes to the Samd14-CP interaction. We affinity purified both HA-Samd14 (bait) observed that the affinity of the Samd14-CP interaction increased in stress erythropoiesis but is not dynamically regulated by erythropoietic stage or activated by SCF stimulation.

3. Given that the main findings were based on PHZ-mediated anemic stress, it would be helpful for the authors to further discuss their findings in the context of more physiologically relevant models of stress erythropoiesis and pathologic settings.

We added text to discussion (Page 22 line 502). “Importantly, while we used PHZ injection as a model for acute anemia, Samd14 expression is also elevated in clinically relevant anemia models induced by severe bleeding and erythroid radioprotection, suggesting that Samd14 mechanisms are not specific to this model.”

Reviewer #2 (Recommendations for the authors):A few more granular comments for the authors:It is not clear from the apoptosis analyses discussed in 238-246 if there was an overall increase in cell death, or only in early/ late apoptosis. Looking at the specific viability of the R2/R3 cells would help to understand if they are maturing at an increased rate, or if they are dying. Also, it's not clear from the data presented that the role of Capzb is limited to maintaining viability in context of anemic stress. Does KD of Capzb in non-stressed erythroblasts or cell lines (like G1E or CD34)+ lead to loss of viability?

As discussed in “essential revisions”, we quantitated viability of cells expressing shRNA against Capzb. This analysis revealed increased numbers of dead (DAPI+) cells at all stages of immunophenotypically defined erythroid cells. Moreover, we also looked at cells isolated from bone marrow, which indicated that Capzb function is not limited to stress erythroid precursor function, since knockdown also increased differentiation and decreased viability in these cells.

For the studies in Figure 5A, it is not clear what cell populations are being studied. Only a fairly narrow window of erythroblasts are typically positive for both Cd71 and kit (CFU-E and ProE), so it seems somewhat counterintuitive to parse this population further by CD71 level.

Prior studies (Flygare, Lodish. 2011, Li, Lodish 2019) describe a continuum from BFU-E to CFU-E using CD71 and Kit as cell surface markers, demonstrating that unique functional states, transcriptional profiles and responses to growth factor stimulation exist throughout this transition. To address this, the revision incorporated functional data highlighting distinctions in erythroid differentiation and colony forming potential between populations of cells in the CD71^low/med/high^. Confirming prior studies, the CD71^med^Kit^+^ population contained a higher fraction of CFU-E compared to CD71^high^Kit^+^ cells.

shRNA can have off target effects in erythroid cells. Use of alternate methods such as TALENs or CRISPR/Cas9 to disrupt Capzb would further support the conclusions that the CP complex has a role in erythropoiesis.It is interesting that the CP components are downregulated following anemic stress, while SAMD14 goes up. How do these expression changes fit into the model of SAMD14-CP interaction in stress erythroid progenitors? Did this change occur on the protein level as well?

While we did not conduct knockout experiments in cells, it is notable that knockdown data is consistent with functional comparisons of full length Samd14 compared to S14-ΔCPB. For example, colony formation is decreased in Capzb knockdown and in comparisons between expression of Samd14 and S14ΔCPB. Knockout experiments would potentially reveal new information, but Capzb knockout may be lethal to cells due the role of barbed end capping in diverse cellular processes (Gerdin et al., Acta Ophthalmologica. 2010). We conducted additional experiments in the revision which confirm that protein levels of Capzb were decreased similarly to mRNA in PHZ-treated spleen. Nothing is currently known about what regulates Capzb expression, but it is expressed at a high level in many cell types. While it is decreasing in anemic stress, the protein remains easily detectable in lin^-^ cells.

Reviewer #3 (Recommendations for the authors):The paper from Ray et al., does an excellent job of identifying new components of the Kit/SCF signaling pathway. The authors have built on previous data showing the Samd14 plays a key role in regulating increased activation of Erk and Akt signaling pathways downstream of Kit receptor during stress erythropoiesis. The CP complex was not previously identified as a component of this pathway. The biochemical analysis is a strength. The weaknesses of the paper have more to do with the phenotypic analysis. The paper does not extend the Samd14 findings that Erk and Akt signaling is affected. It would be greatly improved, if there were mechanistic data showing how the Samd14-CP complex interaction affected Kit signaling. The second weakness is in the analysis of stress erythropoiesis. The data from Samd14 and the knockdown of Capz show a defect in BFU-E. However, the analysis focuses on later stage cells and on cells isolated from spleens after the expansion of the stress BFU-E population has occurred. Furthermore, stress BFU-E respond to different conditions (Epo + BMP4 + SCF at 1% O2) than steady state BFU-E (Epo + IL-3 or SCF), the conditions used here were for steady state BFU-E so some aspects of the role of the CP complex in stress erythropoiesis may be lost. The discussion would also be improved if the role of Samd14/CP complex was put into the developmental context of stress erythropoiesis. The fact that you have fewer BFU-E and when you look at erythroblasts in your cultures, they skew towards more mature cells suggests the defect is in the proliferation of immature cells and the ability of these cells to resist differentiation.

We provide a figure illustrating that Sca1+CD133+Kit+ cells isolated from PHZ-treated spleen include a high degree of autofluorescence, particularly with Sca1 (multiple fluorophores obtained the same result). We do not think that reliable detection of this population from PHZ-treated spleen by flow cytometry is possible.